# RobustZero: Enhancing MuZero Reinforcement Learning Robustness to State Perturbations

**Yushuai Li** [1]  **Hengyu Liu** [1]  **Torben Bach Pedersen** [1]  **Yuqiang He** [2]  **Kim Guldstrand Larsen** [1]  **Lu Chen** [3]
**Christian S. Jensen** [1]  **Jiachen Xu** [1]  **Tianyi Li** [1]

## Abstract

The MuZero reinforcement learning method has achieved superhuman performance at games, and advances that enable MuZero to contend with complex actions now enable use of MuZero-class methods in real-world decision-making applications. However, some real-world applications are susceptible to state perturbations caused by malicious attacks and noisy sensors. To enhance the robustness of MuZero-class methods to state perturbations, we propose RobustZero, the first MuZero-class method that is robust to worst-case and random-case state perturbations, with zero prior knowledge of the environment's dynamics. We present a training framework for RobustZero that features a self-supervised representation network, targeting the generation of a consistent initial hidden state, which is key to obtain consistent policies before and after state perturbations, and it features a unique loss function that facilitates robustness. We present an adaptive adjustment mechanism to enable model update, enhancing robustness to both worst-case and random-case state perturbations. Experiments on two classical control environments, three energy system environments, three transportation environments, and four Mujoco environments demonstrate that RobustZero can outperform state-of-the-art methods at defending against state perturbations.

## 1. Introduction

Deep reinforcement learning (DRL) has achieved notable successes in numerous domains, e.g., energy (Wang et al.,

2022), transportation (Sun et al., 2023), and healthcare (Hao et al., 2022). In DRL, an agent interacts repeatedly with an environment: The agent receives a state update, or an observation, along with a reward; in response, the agent takes an action that in turn yields a new state update and reward. The objective of DRL is to achieve an agent that is able to optimize the cumulative reward. Model-free DRL has the strength of straightforwardly estimating the optimal policy or value that is used to evaluate the policy, without knowing the environment's dynamics. However, model-free DRL suffers from low sample efficiency and lacks sophisticated look-ahead capabilities. Model-based DRL is capable of addressing these weaknesses. However, most model-based DRL methods, e.g., AlphaGo (Silver et al., 2016) and AlphaZero (Silver et al., 2017), require prior knowledge of the environment's dynamics, which restricts their applicability. To relax this requirement, MuZero (Schrittwieser et al., 2020) learns an abstract environment model in combination with Monte Carlo tree search (MCTS). In this sense, MuZero has the strengths of model-free and model-based DRL and therefore has attracted broad attention (Hubert et al., 2021; Ye et al., 2021; Danihelka et al., 2022; Niu et al., 2024; Xuan et al., 2024).

The superhuman performance achieved by MuZero-class methods assumes real and unperturbed states. However, in real-world applications, the observed states may undergo imperceptible perturbations. Even a small "reality gap" can compound errors in the predicted abstract states within MuZero-class methods, causing reduced rewards and potentially harmful decision-making[1]. To date, no studies have been reported that address this problem. To improve robustness, we address two challenges.

***Challenge I: How to achieve robust training for MuZero-class methods under state perturbations?*** The current robust training proposals for DRL focus mainly on designing strategies to: 1) minimize balanced nominal and adversarial losses (Pinto et al., 2017; Zhang et al., 2020a; Oikarinen et al., 2021; Liang et al., 2022); 2) minimize the cross-entropy/KL-divergence policy or value loss between ac-

---

[1]Department of Computer Science, Aalborg University, Aalborg, Denmark [2]School of Computer, Electronics and Information, Guangxi University, Nanning, China [3]College of Computer Science, Zhejiang University, Hangzhou, China. Correspondence to: Tianyi Li <tianyi@cs.aau.dk>, Hengyu Liu <heli@cs.aau.dk>.

*Proceedings of the 42nd International Conference on Machine Learning*, Vancouver, Canada, PMLR 267, 2025. Copyright 2025 by the author(s).

---

[1]Appendix A provides an example to illustrate the importance of considering state perturbations.

tions taken under non-perturbed states and perturbed states (Zhang et al., 2020a; Sun et al., 2022; Zhang et al., 2021; Liang et al., 2022; Zhou et al., 2024; Liu et al., 2024; Dong et al., 2025). These strategies are suitable for DRL methods that use neural networks (NNs) to approximate a policy or value function, such as deep Q-learning (Mnih et al., 2015), deep deterministic policy gradient (DDPG) (Timothy et al., 2015), and proximal policy optimization (PPO) (Engstrom et al., 2020). However, MuZero-class DRL methods differ in that they employ MCTS with a learned model to derive a policy instead of using NNs directly. This calls for a new robust training method to counteract state perturbations.

***Challenge II: How to achieve high robustness for MuZero-class methods to worst-case and random-case state perturbations?*** Worst-case state perturbations yield states that minimize cumulative rewards, while random-case state perturbations imply that perturbations occur randomly in states. Most existing robust DRL methods (Pinto et al., 2017; Zhang et al., 2020a; Oikarinen et al., 2021; Russo & Proutiere, 2021; Liang et al., 2022; Zhang et al., 2021; Sun et al., 2022; Zhou et al., 2024; Zhihe & Xu, 2023; Sun & Zheng, 2024; Dong et al., 2025) investigate defense strategies for worst-case state perturbations, but these methods often suffer from reduced performance when dealing with random-case state perturbations that are prevalent in real-world environments, where sensors produce noisy samples (Zamora et al., 2017). Further, learning a worst-case perturbation policy is complex and challenging (Liu et al., 2024), whereas implementing a random-case perturbation policy is relatively straightforward. To address this issue, a recent study (Liu et al., 2024) proposes an adaptive defense mechanism that achieves high robustness to both worst-case and random-case state perturbations. However, this mechanism cannot be applied to MuZero-class methods because its policy search (Liu et al., 2024) is gradient-based, while MuZero-class methods use gradient-free MCTS for policy search. A new adaptive mechanism is needed for MuZero-class methods.

To tackle these challenges, we propose ***RobustZero***, a novel robust training framework for MuZero-class methods that achieves high robustness to both worst-case and random-case state perturbations. First, by integrating contrastive learning, a self-supervised representation network is developed to generate similar initial hidden states that are mapped from unperturbed states and perturbed states. This enables planning results made in the abstract hidden space with and without perturbations to be consistent. Second, following the self-supervised representation network, we propose an overall loss function that promotes robust training. Third, we design an adaptive adjustment mechanism that initially prioritizes learning from worst-case perturbations and then gradually extends to learn from random-case perturbations on the fly. The adaptive adjustment mechanism enables

learned models to converge to flat minima and escape from steep minima, which further enhances the robustness to unseen worst-case and random-case state perturbations. In summary, our main contributions are as follows.

- We propose RobustZero, a novel robust reinforcement learning framework. We believe this is the first MuZero-class method to defend against state perturbations.

- We propose a self-supervised representation network to facilitate the generation of consistent policies; and based on this, we construct a loss function for robust training.

- We propose a new adaptive adjustment mechanism that enables RobustZero to generate policies that are highly robust to both worst-case and random-case perturbations.

- Extensive experiments on two classical control environments, three energy system environments, three transportation environments, and four Mujoco environments show that RobustZero is able to outperform the state-of-the-art methods in terms of robustness to worst-case and random-case state perturbations.

## 2. Related Work

### 2.1. DRL with MCTS

Many DRL methods have been proposed, primarily using temporal difference learning (Mnih et al., 2015) and gradient-based policy search (Han et al., 2023). In contrast, AlphaGo (Silver et al., 2016), a different DRL method, integrates MCTS as the policy improvement operator with NNs to approximate value and policy networks. Due to the strong look-ahead search capability of MCTS, AlphaGo achieves superhuman performance in numerous games. Subsequently, several DRL methods incorporating MCTS have emerged, including MuZero (Schrittwieser et al., 2020) that can learn abstract environment models without knowledge of underlying dynamics. Diverse MuZero-class methods have been proposed, addressing aspects such as limited action access (Danihelka et al., 2022), complex action spaces (Hubert et al., 2021), limited data (Ye et al., 2021), and time consumption (Xuan et al., 2024). Of these, sampled MuZero (S-MuZero) (Hubert et al., 2021) supports complex action spaces, which is important in real-world decision-making. We extend S-MuZero to achieve robustness to state perturbations, an aspect overlooked by existing MuZero-class methods.

### 2.2. Robust Reinforcement Learning

DRL robustness may relate to multiple aspects, e.g., action perturbations (Tessler et al., 2019; Bukharin et al., 2024), state perturbations (Pinto et al., 2017; Zhang et al., 2020a; Oikarinen et al., 2021; Russo & Proutiere, 2021; Zhang et al., 2021; Sun et al., 2022; Liang et al., 2022; Zhou et al., 2024; Liu et al., 2024; Zhihe & Xu, 2023; Sun & Zheng,

2024; Dong et al., 2025), reward corruptions (Zhang et al., 2020b; Eysenbach & Levine, 2021), and environment discrepancies (Sinha et al., 2020; Huang et al., 2022). We target DRL robustness to state perturbations. A robust adversarial reinforcement learning method is proposed (Pinto et al., 2017) to jointly train an agent and an adversary, where the agent aims to accomplish the primary task objectives while learning to remain robust against disturbances introduced by the adversary. Zhang et al. (Zhang et al., 2020a) offer a foundational theory that equates learning an optimal adversary to finding an optimal policy within a new Markov Decision Process (MDP). They propose a principled policy regularization method to defend against state perturbations. Following this work, the alternating training with learned adversaries (ATLA)-PPO (Zhang et al., 2021) and the Policy Adversarial (PA)-ATLA-PPO (Sun et al., 2022) are designed for launching worst-case perturbation policies under black-box and white-box attack scenarios, respectively. With a similar idea of formulating the design of an optimal adversary as the solution to another MDP, a black-box attack strategy based on DDPG (BA-DDPG) is proposed (Russo & Proutiere, 2021) to derive the worst-case perturbation policy. We consider black-box attacks, where the adversary has no knowledge of the policy and model of the agent. A recent study (Dong et al., 2025) proposes the use of variational optimization over worst-case adversary distributions, rather than a single adversary, and trains an agent to maximize the lower quantile of returns to mitigate over-optimism. As a different branch, several studies (Zhihe & Xu, 2023; Sun & Zheng, 2024) introduce the diffusion models to enhance robustness to state perturbations. Specially, a diffusion model-based predictor is proposed (Zhihe & Xu, 2023) for offline RL to recover the actual states against state perturbations. A belief-enriched pessimistic Q-learning method is proposed (Sun & Zheng, 2024) by using diffusion model to purify observed states. We note that the studies covered here (Pinto et al., 2017; Zhang et al., 2020a; Oikarinen et al., 2021; Russo & Proutiere, 2021; Liang et al., 2022; Zhang et al., 2021; Sun et al., 2022; Zhou et al., 2024; Zhihe & Xu, 2023; Sun & Zheng, 2024; Dong et al., 2025) target worst-case state perturbations. Recently, the most related work (Liu et al., 2024) proposes an new method, called PROTECTED, that outperforms the state-of-the-art methods under both worst-case and random-case state perturbations. However, as discussed above, this method employs gradient-based policy search and is inapplicable in our setting.

### 2.3. Contrastive Representation Learning

The contrastive representation learning has been used to improve the sample efficiency (Schwarzer et al., 2021; Ye et al., 2021) and enhance the representation ability of states (Nasiriany et al., 2019), rewards (Kang et al., 2023), and value functions (Eysenbach et al., 2022). Among them,

one notable work is EfficientZero (Ye et al., 2021), which significantly improves the sample efficiency of the MuZero method while maintaining superior performance. It achieves this by using contrastive representation learning to build a consistent environment model and by using the learned model to correct off-policy value targets. Different from these studies, we aim to leverage contrastive representation learning to improve the robustness of MuZero-class methods to state perturbations.

## 3. Preliminaries

### 3.1. Markov Decision Process (MDP)

**Definition 3.1.** An MDP is a 4-tuple $\mathcal{M} = (\mathcal{S}, \mathcal{A}, \mathcal{P}, \mathcal{R}, \rho_0)$, where $\mathcal{S}$ is a state space, $\mathcal{A}$ is an action space, $\mathcal{P}$ is a deterministic transition function, $\mathcal{R}$ is a reward function, and $\rho_0$ is the initial state distribution.

An MDP is used widely to describe and model an environment in DRL. An agent observes a current state $s_t \in \mathcal{S}$. Driven by a policy, the agent takes an action $a_{t+1} \in \mathcal{A}$ and then receives an updated state $s_{t+1}$ and a reward $r_{t+1} = \mathcal{R}(s_t, a_{t+1})$. For a sequential decision-making task (e.g., voltage control), the goal is to maximize the expected return $\sum_{t=0}^{T} \gamma_t r_t$, where $\gamma_t$ is the discount factor at time $t$.

### 3.2. MuZero

MuZero learns an abstract MDP to represent an abstract environment model and employs MCTS for planning. Note that planning occurs in the abstract MDP based on abstract states instead of real states. To avoid the misuse of symbols, we use $t$ to represent the time step in the real world. Planning is made at each time step $t$. We use $k = 0, ..., K$ to represent the index of unrolled steps in the abstract MDP. Next, we summarize the components of the abstract MDP: 1) a representation network $\mathcal{H}_\theta$ that maps a real state $o_t$ to an abstract and initial state $s_t^0$, denoted as $s_t^0 := \mathcal{H}_\theta(o_t)$; 2) a dynamics network $\mathcal{G}_\theta$ that maps a previous abstract state $s_t^{k-1}$ and a candidate action $a_t^k$ to an immediate reward $r_t^k$ and a new abstract state $s_t^k$, denoted as $r_t^k, s_t^k := \mathcal{G}_\theta(s_t^{k-1}, a_t^k)$; and 3) a prediction network $\mathcal{F}_\theta$ that maps $s_t^k$ to a policy $p_t^k$ and a value function $v_t^k$, denoted as $p_t^k, v_t^k := \mathcal{F}_\theta(s_t^k)$, where $\theta$ denotes the NN parameters of the representation, dynamics, and prediction networks. By using the abstract MDP, MCTS obtains a search policy $\pi_t$ at each time step $t$ to determine the real action $a_{t+1}$. Then, the real environment receives $a_{t+1}$ and further provides a new state $o_{t+1}$ and a real reward $\mu_{t+1}$. By employing value equivalence, the training objective of MuZero is to enable achieving $p_t^k \approx \pi_{t+k}$, $v_t^k \approx z_{t+k}$, and $r_t^k \approx \mu_{t+k}$, where $z_{t+k}$ is the sampled return. The sequence of real actions from the sampled trajectory is used to compute $r_t^k$. In this way, the planning results made in the abstract MDP are equivalent to those in the real environment.

### 3.3. S-MuZero

The traditional MuZero method is only suitable for low-dimensional and discrete action spaces. By combing the sample strategy and MCTS, S-MuZero (Hubert et al., 2021) enables planning in complex action spaces, including continuous and high-dimensional discrete spaces. For ease of distinction, we name the policy evaluation and improvement in S-MuZero as *sampled MCTS*, denoted as $s\text{-}MCTS(s_t^0, \theta, \mathcal{K})$, where $\mathcal{K}$ is the number of sampled actions. We employ sampled MCTS to obtain the search policy $\pi_t \leftarrow s\text{-}MCTS(s_t^0, \theta, \mathcal{K})$, where $\pi_t$ is also called the *sample-based improved policy* (Hubert et al., 2021).

## 4. Problem Statement

### 4.1. State Perturbations

An adversary or attacker aims to launch state perturbations to reduce the accumulated reward gained by an agent. We consider the black-box attack and assume that the adversary cannot change the MDP. The adversary can then only observe and perturb states before they reach the agent. To obtain general robust-defense performance, we consider both worst-case and random-case state perturbations. Next, each perturbed state is subject to a budget constraint (Oikarinen et al., 2021; Liang et al., 2022; Zhou et al., 2024; Zhang et al., 2020a; 2021; Liu et al., 2024; Sun et al., 2022), which limits the power of the adversary.

**Definition 4.1.** The **budget constraint** $B_\epsilon(o_t)$ denotes an $l_p$ norm ball centered at $o_t$ with radius $\epsilon$.

For clarity, we use superscripts "~" and "^" to represent the relevant symbols and variables under worst-case and random-case state perturbations, respectively. For instance, $\widetilde{t}$ and $\hat{t}$ denote data at time $t$ from a trajectory generated under worst-case and random perturbations, respectively, and $\widetilde{o_{\widetilde{t}}}$ and $\hat{o}_{\hat{t}}$ denote the perturbed states after worst-case and random-case state perturbations, respectively. Next, we formalize the worst-case and random-case perturbation policies.

**Definition 4.2.** A **worst-case perturbation policy** is defined by a function $\mathbb{A}_{\text{worst}}$ that maps $o_{\widetilde{t}}$ to $\widetilde{o_{\widetilde{t}}} := \mathbb{A}_{\text{worst}}(o_{\widetilde{t}}) \in \mathcal{B}_\epsilon(o_{\widetilde{t}})$, with the objective of finding an optimal policy $\mathbb{A}_{\text{worst}}^*$ that minimizes $\sum_{\widetilde{t}=0}^{T} \gamma_{\widetilde{t}} \widetilde{\mu}_{\widetilde{t}}$.

**Definition 4.3.** A **random perturbation policy** is defined by a function $\mathbb{A}_{\text{random}}$ that maps $o_{\hat{t}}$ to a random state $\hat{o}_{\hat{t}} := \mathbb{A}_{\text{random}}(o_{\hat{t}}) \in \mathcal{B}_\epsilon(o_{\hat{t}})$.

### 4.2. Objective of RobustZero

Similar to MuZero, RobustZero aims to learn an abstract MDP that, by employing MCTS, produces a planned policy, reward, and value. These planning results are capable of aligning with those of the policy, reward, and value in the real environment, without relying on the actual dynamics of the environment. Unlike MuZero, the additional and unique objective of RobustZero is to generate consistent planning results both without and with state perturbations restricted by a budget constraint. This aims to enable satisfactory planning results even when the agent receives perturbed states, resulting in enhanced robustness of the method.

## 5. RobustZero

The RobustZero framework is shown in Fig. 1. It encompasses three parts: parallel data collection under state perturbations, self-supervised representation network and loss function, and adaptive adjustment during training.

### 5.1. Parallel Data Collection under State Perturbations

To increase the efficiency of data collection, RobustZero operates in two parallel environments, Environment 1 and Environment 2, under worst-case and random-case state perturbations, respectively. Thus, two replay buffers, worst-case buffer $\mathcal{D}_{WC}$ and random-case buffer $\mathcal{D}_{RC}$, are used to store different data. The details of the parallel data collection are provided in Appendix B.1.

### 5.2. Self-supervised Representation Network and Loss Function

To obtain similar planning results with and without state perturbations, we propose a self-supervised representation network that replaces MuZero's original representation network. Then, we define a new loss function—encompassing a worst-case loss term, a random-case loss term, and a decay term—to guide the training of the learned model.

a) **Model of self-supervised representation network**: Recalling the structure of MuZero-like methods, the representation network serves as the interface between the real-world states and the abstract hidden states. Taking states before and after perturbations, i.e., $o_{\widetilde{t}}$ and $\widetilde{o_{\widetilde{t}}}$, as input to the representation network, we can get the corresponding roots of the search trees, i.e., $s_{\widetilde{t}}^0 \leftarrow \mathcal{H}_\theta(o_{\widetilde{t}})$ and $\widetilde{s}_{\widetilde{t}}^0 \leftarrow \mathcal{H}_\theta(\widetilde{o_{\widetilde{t}}})$. If the encoded root nodes are the same, the final policies generated by the sampled MCTS will also remain the same. In this sense, the agent is capable of defending against state perturbations well.

With this inspiration, we include a self-supervised representation network that leverages contrastive learning in the framework. The network contains two branches. The first branch contains a representation network and a projector network without gradients. The second branch contains the same representation network, the same projector network, and an additional predictor network. We use $\phi$ to represent the NN parameters of the projector and the predictor. Next, we define the notions of projector and predictor.

**Definition 5.1.** A projector $P1_\phi$ transforms an initial hidden

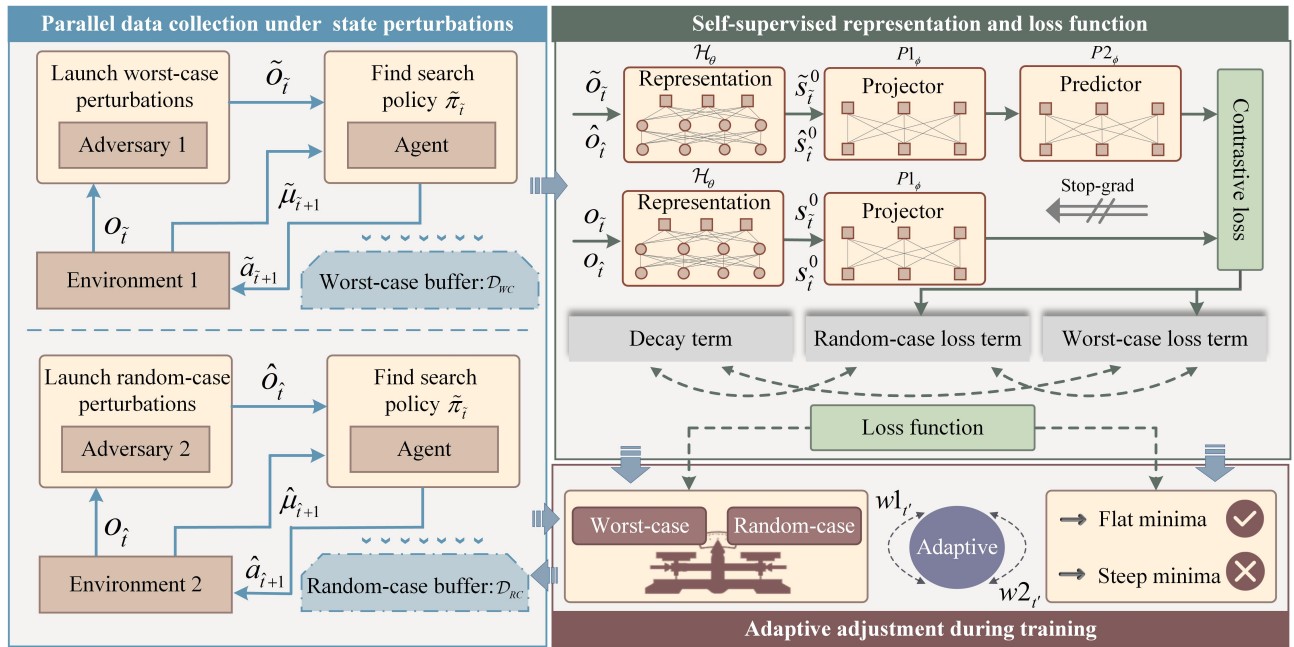

*Figure 1.* Overview of the RobustZero framework.

state to a d-dimensional vector.

**Definition 5.2.** A predictor $P2_\phi$ transforms the output of the projector to another d-dimensional vector.

The key functionality of the projector is to map high-dimensional features from the encoder to a lower-dimensional space, making the learned features more suitable for the subsequent contrastive loss computation. Next, the key functionality of the predictor is to transform projected features to stabilize optimization and reduce collapsing solutions. Specifically, if we omit the predictor and directly enforce similarity between the outputs of two branches (e.g., using mean squared error or cosine similarity), the model may exploit a shortcut by collapsing to a constant output vector. In such a scenario, since all inputs produce the same output, the loss is minimized regardless of input variation, resulting in model collapse. When collapse occurs, all states—regardless of whether they are perturbed or not—are mapped to the same or very similar initial hidden state, effectively losing the ability to distinguish between different inputs. To avoid model collapse, one effective strategy is to introduce an asymmetric network architecture, where a predictor is added to one branch, while the other branch remains without it and has its gradient flow stopped. This asymmetry prevents both branches from trivially converging to the same constant output. The branch with the predictor learns to align its output with the target branch, which acts as a stable reference point. Because the target branch does not receive gradients, it cannot adjust itself to match the predictor's potentially trivial solution, thus breaking the symmetry that often leads to collapse. With this model, we can supervise the encoded root node after

perturbation by using the encoded root node before perturbation to improve the representation network. The goal is to achieve maximum similarity between the two branches. This enables the improved representation network to pull $\widetilde{s}_{\widetilde{t}}^0$ ($\hat{s}_{\hat{t}}^0$) and $s_{\widetilde{t}}^0$ ($s_{\hat{t}}^0$) close to each other, resulting in enhanced robustness to state perturbations.

b) **Contrastive loss**: To achieve the objective of the self-supervised representation network, we define worst-case and random-case contrastive loss functions:

$$\widetilde{\mathcal{L}}_{ctr} = \mathbb{E}_{\mathcal{B} \sim \mathcal{D}_{WC}} \left[ -\frac{P2_\phi(P1_\phi(\widetilde{s}_{\widetilde{t}}^0))}{\|P2_\phi(P1_\phi(\widetilde{s}_{\widetilde{t}}^0))\|_2} \cdot \frac{sg(P1_\phi(s_{\widetilde{t}}^0))}{\|sg(P1_\phi(s_{\widetilde{t}}^0))\|_2} \right] \tag{1}$$

$$\hat{\mathcal{L}}_{ctr} = \mathbb{E}_{\mathcal{B} \sim \mathcal{D}_{RC}} \left[ -\frac{P2_\phi(P1_\phi(\hat{s}_{\hat{t}}^0))}{\|P2_\phi(P1_\phi(\hat{s}_{\hat{t}}^0))\|_2} \cdot \frac{sg(P1_\phi(s_{\hat{t}}^0))}{\|sg(P1_\phi(s_{\hat{t}}^0))\|_2} \right], \tag{2}$$

where $\| \cdot \|_2$ is the $l_2$-norm; $sg(\cdot)$ is the stop-gradient operation; and $\mathcal{B}$ is the batch.

c) **Overall loss function**: The loss function of RobustZero has three weighted terms: i) a worst-case loss term, ii) a random-case loss term, and iii) a decay term. The first of these is composed of the original loss function and the contrastive loss function:

$$\widetilde{\mathcal{L}} = \mathbb{E}_{\mathcal{D}_{WC} \sim \mathcal{B}} \left[ \sum_{k=1}^{K} \widetilde{l}^r(\widetilde{\mu}_{\widetilde{t}+k}, \widetilde{r}_{\widetilde{t}}^k) + c_1 \sum_{k=0}^{K} KL^p(\widetilde{\pi}_{\widetilde{t}+k}, \widetilde{p}_{\widetilde{t}}^k) \right.$$
$$\left. + c_2 \sum_{k=0}^{K} \widetilde{l}^v(\widetilde{z}_{\widetilde{t}+k}, \widetilde{v}_{\widetilde{t}}^k) \right] + c_3 \widetilde{\mathcal{L}}_{ctr}, \tag{3}$$

where $c1$, $c2$, and $c3$ are loss coefficients; $\widetilde{l^r}$ and $\widetilde{l^v}$ are the Cross-Entropy loss functions for reward and value, respectively; and $KL^p$ is the KL-divergence loss function for policy.

The settings of $\widetilde{l^r}$, $KL^p$, and $\widetilde{l^v}$ are the same as their counterparts in S-MuZero, with the objectives of minimizing the errors between the predicted policy $\widetilde{p}_{\widetilde{t}}^k$ and the real search policy $\widetilde{\pi}_{\widetilde{t}+k}$, between the predicted value $\widetilde{v}_{\widetilde{t}}^k$ and the target value $\widetilde{z}_{\widetilde{t}+k}$, and between the predicted reward $\widetilde{r}_{\widetilde{t}}^k$ and the real reward $\widetilde{\mu}_{\widetilde{t}+k}$.

Similarly, the random-case loss term is defined as:

$$\hat{\mathcal{L}} = \mathbb{E}_{\mathcal{D}_{RC}\sim\mathcal{B}}\left[\sum_{k=1}^{K}\hat{l}^r(\hat{\mu}_{\hat{t}+k}, \hat{r}_{\hat{t}}^k) + c_1\sum_{k=0}^{K}KL^p(\hat{\pi}_{\hat{t}+k}, \hat{p}_{\hat{t}}^k)\right.$$
$$\left. + c_2\sum_{k=0}^{K}\hat{l}^v(\hat{z}_{\hat{t}+k}, \hat{v}_{\hat{t}}^k)\right] + c_3\hat{\mathcal{L}}_{ctr} \quad (4)$$

As a regularizer, the decay term is added to the main loss function to avoid over-fitting:

$$\mathcal{L}_{decay} = \|\theta\|_2^2 \quad (5)$$

With these components, we define RobustZero's loss function as:

$$\mathcal{L} = \widetilde{\mathcal{L}} + w1_{t'}\hat{\mathcal{L}} + \frac{w2_{t'}}{2}\mathcal{L}_{decay}, \quad (6)$$

where $w1_{t'}$ and $w2_{t'}$ are weights of the loss function, and $t'$ is the time index in the training phase. The performance of RobustZero is affected by the settings of the two weights. We thus consider shortly how to adjust $w1_{t'}$ and $w2_{t'}$ dynamically to enhance RobustZero's robustness to both worst-case and random-case state perturbations.

### 5.3. Adaptive Adjustment during Training

We proceed to explain how to train RobustZero. We recall that the worst-case and random-case perturbation policies reflect the worst and average scenarios, respectively. Although both types of perturbations are important, we hope to first increase the weight of the worst-case loss term to improve the robustness of RobustZero to worst-case perturbations, and then gradually accommodate the average-case during the training phase. The main reasons are: 1) the worst-case perturbation policy is purposeful and smart, while the random-case perturbation policy is random. Thus, first improving the ability of the model to cope with the worst-case perturbation is beneficial for the stability of the overall training process; and 2) worst-case state perturbations are more dangerous than random-case state perturbations. This makes it preferable to weight the worst-case highly. To achieve this, we design $w1_{t'}$ as:

$$w1_{t'} = 2\cdot sg\left(\frac{1}{1+e^{\lambda_1(\widetilde{\mathcal{L}}_{ctr}+1)}}\right), \quad (7)$$

where $\lambda_1$ is a hyperparameter that affects the rate of change of $w1_{t'}$.

Since $\widetilde{\mathcal{L}}_{ctr} \in [-1, 1]$, we have $\widetilde{\mathcal{L}}_{ctr} + 1 \in [0, 2]$. The smaller $\widetilde{\mathcal{L}}_{ctr}$ is, the more similar $\widetilde{s}_{\tilde{t}}^0$ and $s_{\tilde{t}}^0$ are, indicating that RobustZero is more robust to worst-case state perturbations. According to Eq. 7, $w1_{t'}$ increases as $\widetilde{\mathcal{L}}_{ctr}$ decreases. This feature allows us to first assign a high weight to the worst-case loss term, enabling updates to $\theta$ to strengthen the model's robustness to worst-case perturbations. Then, as the similarity between $\widetilde{s}_{\tilde{t}}^0$ and $s_{\tilde{t}}^0$ increases, we gradually adjust $w1_{t'}$ to allow RobustZero to accommodate random-case state perturbations. When $\widetilde{\mathcal{L}}_{ctr} = -1$, we have $w1_{t'} = 1$. In this case, the worst-case and random-case loss terms have equal weight, which occurs only if the trained RobustZero model can handle worst-case state perturbations fully.

Next, we design $w2_{t'}$. Note that while training guided by the combination of the worst-case and random-case loss terms can enhance the robustness of RobustZero, the trained model may experience compromised performance in the presence of unseen worst-case or random-case state perturbations. To address this issue, we dynamically adjust the decay term to guide the learned model towards flat minima and away from steep minima, thereby improving generalization. To this end, we employ the gradient information (Ghiasi et al., 2023) to determine the dynamic setting of $w2_{t'}$. More specifically, the gradient update for parameter $\theta$ from time step $t'-1$ to $t'$ is:

$$\theta_{t'} = \theta_{t'-1}$$
$$- \varsigma\left(\nabla\widetilde{\mathcal{L}}_{\theta_{t'-1}} + w1_{t'}\nabla\hat{\mathcal{L}}_{\theta_{t'-1}} + w2_{t'}\theta_{t'-1}\right), \quad (8)$$

where $\varsigma$ is the stepsize and $\nabla\widetilde{\mathcal{L}}_{\theta_{t'-1}}$ and $\hat{\mathcal{L}}_{\theta_{t'-1}}$ are gradients related to $\theta$ at $t'-1$ from Eqs. 3 and 4, respectively. Similarly, we let $\nabla\widetilde{\mathcal{L}}_{\phi_{t'-1}}$ and $\hat{\mathcal{L}}_{\phi_{t'-1}}$ denote the gradients related to $\phi$ at $t'-1$ from Eqs. 3 and 4, respectively.

In Eq. 8, items $\nabla\widetilde{\mathcal{L}}_{\theta_{t'-1}} + w1_{t'}\nabla\hat{\mathcal{L}}_{\theta_{t'-1}}$ and $w2_{t'}\theta_{t'-1}$ quantify the actual strength of an update of $\theta$. Further, $\lambda_2$ is used to capture the ratio between the magnitudes of $w2_{t'}\theta_{t'-1}$ and $\nabla\widetilde{\mathcal{L}}_{\theta_{t'-1}} + w1_{t'}\nabla\hat{\mathcal{L}}_{\theta_{t'-1}}$:

$$\lambda_2 = \frac{\|w2_{t'}\theta_{t'-1}\|_2}{\|\nabla\widetilde{\mathcal{L}}_{\theta_{t'-1}} + w1_{t'}\nabla\hat{\mathcal{L}}_{\theta_{t'-1}}\|_2} \quad (9)$$

In order to maintain the actual effect of the main loss (i.e., $\widetilde{\mathcal{L}} + w1_{t'}\hat{\mathcal{L}}$) and the decay unchanged, we keep $\lambda_2$ as a constant. This enables us to calculate $w2_{t'}$ as:

$$w2_{t'} = sg\left(\frac{\lambda_2\|\nabla\widetilde{\mathcal{L}}_{\theta_{t'-1}} + w1_{t'}\nabla\hat{\mathcal{L}}_{\theta_{t'-1}}\|_2}{\|\theta_{t'-1}\|_2}\right) \quad (10)$$

Eq. 10 shows that $w2_{t'}$ is updated adaptively according to the changes to $\nabla\widetilde{\mathcal{L}}_{\theta_{t'-1}}$, $\nabla\hat{\mathcal{L}}_{\theta_{t'-1}}$, $w1_{t'}$, and $\theta_{t'-1}$. Then,

the weighted decay term in Eq. 6 can be re-written as:

$$\frac{w2_{t'}}{2}\mathcal{L}_{decay} = \frac{sg(\lambda_2 \|\nabla\widetilde{\mathcal{L}}_{\theta_{t'-1}} + w1_{t'}\nabla\hat{\mathcal{L}}_{\theta_{t'-1}}\|_2)\|\theta_{t'-1}\|_2}{2} \tag{11}$$

Eq. 11 indicates that the weighted decay term increases with the gradient of the main loss function, and vice versa. The incorporation of gradient information brings two major benefits. First, when the gradient of the main loss function approaches zero, the weight decay term also approaches zero. This prevents over-optimization of the weighted decay term in flat minima, allowing stronger emphasis on the main loss function, which is beneficial for obtaining a model that is more robust to state perturbations. Second, a large weighted decay term is applied when the gradient of the main loss function is large. This prevents the model from settling into steep local minima, thereby reducing the risk of early overfitting during training. In this sense, adaptive adjustment of $w2_{t'}$ enables the learned model to converge to flat minima and escape from steep minima.

Next, to achieve more stable training, we use smooth update rules for both $w1_{t'}$ and $w2_{t'}$, i.e.:

$$w1_{t'} \leftarrow 0.1w1_{t'-1} + 0.9w1_{t'} \tag{12}$$

$$w2_{t'} \leftarrow 0.1w2_{t'-1} + 0.9w2_{t'} \tag{13}$$

The pseudocode of the training process to update the model of RobustZero is provided in Appendix B.2.

## 6. Experiments

### 6.1. Experimental Setup and Baselines

We study RobustZero on: 1) two classical control environments, including CartPole[2] and Pendulum[3]; 2) three energy environments in power distribution systems with hybrid action spaces (Fan et al., 2022), including IEEE 34-bus, IEEE 123-bus, and IEEE 8500-node systems; 3) three transportation environments [4], including Highway with discrete action space, Intersection with discrete action space, and Racetrack with continuous action space; and 4) four Mujoco environments with continuous action spaces, including Hopper, Walker2d, HalfCheetah, and Ant, following a setup similar to that of Liu et al. (2024). Therein, the three energy environments support the testing of voltage control tasks with the objective of minimizing the total cost of voltage violations, control errors, and power losses, while meeting both networked and device constraints. The action spaces of IEEE 34-bus, IEEE 123-bus, and IEEE 8500-node are 10-dimensional (8 continuous actions and 2 discrete actions),

15-dimensional (11 continuous actions and 4 discrete actions), and 32-dimensional (22 continuous actions and 10 discrete actions), respectively. The three transportation environments support the testing of autonomous driving tasks. Therein, an autonomous driving car interacts with other vehicles to navigate different scenarios: i) Highway−Drive fast, avoid collisions, and stay in the right-most lane; ii) Intersection−Cross safely, follow traffic rules, and keep a steady speed; and iii) Racetrack−Finish quickly while staying on track and driving smoothly. The action space of an autonomous driving car is two-dimensional. Furthermore, we compare with five baselines:

- ATLA-PPO (Zhang et al., 2021) can obtain worst-case perturbation policies under black-box attacks and can defend against such attacks. ATLA-PPO is trained under worst-case state perturbations.
- PROTECTED (Liu et al., 2024) is the state-of-the-art and most related model-free DRL method for handling worst-case and random-case state perturbations. PROTECTED is trained under worst-case and random-case state perturbations.
- S-MuZero (Hubert et al., 2021) is the most related MuZero-class method supporting complex action spaces. S-MuZero is unable to defend against state perturbations and is trained under no state perturbation.
- S-MuZero-worst and S-MuZero-random are baselines, representing S-MuZero trained under worst-case and random-case state perturbations, respectively.

Following Zhang et al. (2021) and Liu et al. (2024), we adopt ATLA-PPO to obtain $\mathbb{A}_{worst}$ that reflects the strongest impact on reducing the accumulated reward. The Uniform noise is used to execute $\mathbb{A}_{random}$ that reflects the average impacts of all possible perturbations. Details of the experimental settings are provided in Appendix C.1.

### 6.2. Comparison Study

We conduct the main experiments to assess robustness of RobustZero. We evaluate the performance of the methods under three scenarios: i) no perturbations, ii) worst-case state perturbations, and iii) random-case state perturbations. The average episodic rewards $\pm$ the standard deviation over 50 episodes for the three scenarios (natural, worst-case, and random-case) are reported in Table 1. In each column, the highest reward is highlighted in bold. Due to the space limitation, Table 1 only reports the results on Pendulum, IEEEE 34-bus, IEEE 8500-node, and Racetrack. The results and analyses for the remaining environments are provided in Appendix C.2. Overall, RobustZero outperforms all five baselines at defending against state perturbations across the four environments. The details are as follows.

- RobustZero and S-MuZero achieve similar natural rewards that are higher than those of the other baselines.

---

[2] gymnasium.farama.org/environments/classic_control/cart_pole/
[3] gymnasium.farama.org/environments/classic_control/pendulum/
[4] github.com/Farama-Foundation/HighwayEnv

Table 1. Main experimental results on Pendulum, IEEE 34-bus, IEEE 8500-node, and Racetrack.

| Method | Pendulum | | | IEEE 34-bus | | |
|---|---|---|---|---|---|---|
| | Natural Reward | Worst-case Reward | Random-case Reward | Natural Reward | Worst-case Reward | Random-case Reward |
| ATLA-PPO | -90.26 ± 64.32 | -94.24 ± 66.42 | -92.46 ± 69.42 | -11.36 ± 0.12 | -14.07 ± 0.33 | -13.56 ± 0.17 |
| PROTECTED | -89.46 ± 59.83 | -91.96 ± 51.24 | -91.32 ± 55.44 | -10.43 ± 0.09 | -12.89 ± 0.32 | -12.26 ± 0.07 |
| S-MuZero | **-82.58 ± 57.05** | -250.02 ± 102.56 | -166.89 ± 62.53 | **-8.56 ± 0.19** | -25.70 ± 6.41 | -19.50 ± 2.95 |
| S-MuZero-worst | -251.49 ± 2.89 | -89.98 ± 48.36 | -170.57 ± 68.96 | -15.31 ± 0.47 | -15.07 ± 0.39 | -15.79 ± 2.59 |
| S-MuZero-random | -88.91 ± 56.90 | -168.70 ± 60.55 | -130.29 ± 99.12 | -10.96 ± 0.49 | -15.76 ± 2.19 | -14.68 ± 0.39 |
| **RobustZero** | -83.48 ± 58.96 | **-87.92 ± 53.24** | **-84.68 ± 55.74** | -9.57 ± 0.43 | **-12.16 ± 1.19** | **-11.43 ± 0.37** |

| Method | IEEE 8500-node | | | Racetrack | | |
|---|---|---|---|---|---|---|
| | Natural Reward | Worst-case Reward | Random-case Reward | Natural Reward | Worst-case Reward | Random-case Reward |
| ATLA-PPO | -1279 ± 9 | -1412 ± 12 | -1351 ± 7 | 409.13 ± 0.31 | 386.69 ± 0.16 | 392.87 ± 0.01 |
| PROTECTED | -1264 ± 4 | -1376 ± 19 | -1296 ± 19 | 478.40 ± 3.00 | 429.36 ± 5.14 | 459.10 ± 6.03 |
| S-MuZero | **-1135 ± 18** | -1863 ± 25 | -1686 ± 15 | **567.80 ± 3.16** | 352.20 ± 6.93 | 370.40 ± 3.94 |
| S-MuZero-worst | -1402 ± 26 | -1457 ± 21 | -1521 ± 10 | 375.9 ± 6.32 | 410.00 ± 6.94 | 398.70 ± 6.32 |
| S-MuZero-random | -1294 ± 20 | -1503 ± 18 | -1368 ± 5 | 415.60 ± 4.64 | 390.50 ± 7.64 | 410.82 ± 6.06 |
| **RobustZero** | -1158 ± 30 | **-1374 ± 35** | **-1217 ± 27** | 520.60 ± 1.32 | **477.00 ± 1.26** | **494.00 ± 2.42** |

The major reasons are: i) MuZero-class methods use an abstract environment model to make plan ahead, which enables better solutions than model-free DRL, e.g., ATLA-PPO and PROTECTED; and ii) S-MuZero-worst and S-MuZero-random cannot obtain comparable natural rewards because they are trained under worst-case state perturbations only and random-case state perturbations only, respectively. Additionally, S-MuZero obtains slightly higher natural rewards than RobustZero. This is because S-MuZero is trained without state perturbations. It thus obtains the best natural rewards. However, its worst-case and random-case rewards decrease notably. In comparison, RobustZero can still obtain comparable natural rewards but much better in both worst-case and random-case rewards.

- RobustZero achieves a higher worst-case reward than all baselines across the four environments. RobustZero increases the worst-case reward by 2.34%–184.37% on Pendulum, by 6.00%–111.35% on IEEE 34-bus, by 0.15%–35.59% on IEEE 8500-node, and by 11.10%–35.43% on Racetrack. Further, S-MuZero-worst obtains a higher worst-case reward than S-MuZero and S-MuZero-random. Notably, its worst-case rewards appear higher than the corresponding random-case rewards. This is because S-MuZero-worst does not incorporate any robustness mechanism. It is trained solely on perturbed states, without any information about non-perturbed states. This causes S-MuZero-worst to be over-trained under the worst-case perturbation policy, making it highly specialized and adapted to that particular perturbation pattern. As a result, its performance under worst-case state perturbations unusually appears higher than its performance under random-case state perturbations as observed in Table 1. Although S-MuZero-worst achieves relatively high worst-case rewards, its natural reward and random-case reward are low, due to the lack of a defense strategy.

- RobustZero achieves a higher random-case reward than all baselines across all the four environments, increasing this reward by 7.84%–101.43% on Pendulum, by 7.26%–70.60% on IEEE 34-bus, by 6.49%–38.54% on IEEE 8500-node, and by 7.60%–33.37% on Racetrack. Similarly, S-MuZero-random achieves a high random-case reward, but also reduced natural and worst-case rewards due to the absence of a defense strategy.

We make additional comparisons: 1) Evaluation of all methods under perturbation policies that were not encountered during training, to assess the generalization capability of RobustZero (see Appendix C.5); 2) Time and sampling efficiency analysis that focuses on the training and sampling time, and sampling efficiency (see Appendix C.6).

### 6.3. Ablation Study

We study the effects of the self-supervised representation network, the adaptive adjustment mechanism, and the parallel use of worst-case and random-case buffers on improving the robustness of RobustZero. We use RobustZero/cl to denote RobustZero without the self-supervised representation network, use RobustZero/$w1$ to denote RobustZero trained with $w1 = 1$[5], use RobustZero/$w2$ to denote RobustZero trained with $w2$ set to 5e-6, 5e-8, 5e-8, and 1e-5 for Pendulum, IEEEE 34-bus, IEEE 8500-node, and Racetrack, respectively[6], use RobustZero-$\mathcal{D}_{WC}$ to denote RobustZero employing only the worst-case buffer, and use RobustZero-$\mathcal{D}_{RC}$ to denote RobustZero employing only the random-case buffer. Similarly, we report results for Pendulum, IEEEE 34-bus, IEEE 8500-node, and Racetrack, as shown in Table 2. The counterparts for the remaining environments are

---

[5]$w1 = 1$ indicates that worst-case and random-case perturbations are treated equally.

[6]The method for selecting these values of $w2$ is provided in Appendix C.3.

*Table 2.* Ablation study on Pendulum, IEEE 34-bus, IEEE 8500-node, and Racetrack.

| Method | Pendulum | | | IEEE 34-bus | | |
|---|---|---|---|---|---|---|
| | Natural Reward | Worst-case Reward | Random-case Reward | Natural Reward | Worst-case Reward | Random-case Reward |
| RobustZero/cl | -85.43 ± 54.00 | -91.15 ± 58.15 | -87.24 ± 55.00 | -12.03 ± 0.30 | -15.18 ± 1.15 | -14.03 ± 0.32 |
| RobustZero/$w1$ | -83.99 ± 54.64 | -89.41 ± 49.42 | -85.19 ± 55.19 | -10.12 ± 0.42 | -13.41 ± 1.30 | -12.20 ± 0.26 |
| RobustZero/$w2$ | -84.67 ± 56.36 | -90.23 ± 57.00 | -86.84 ± 55.24 | -11.57 ± 0.36 | -14.63 ± 1.10 | -13.49 ± 0.24 |
| RobustZero-$\mathcal{D}_{WC}$ | -89.36 ± 56.10 | -90.63 ± 58.02 | -90.88 ± 54.69 | -14.16 ± 0.50 | -14.89 ± 0.71 | -14.76 ± 0.40 |
| RobustZero-$\mathcal{D}_{RC}$ | -87.67 ± 55.81 | -92.26 ± 57.66 | -89.09 ± 56.02 | -13.60 ± 0.61 | -14.96 ± 1.06 | -14.31 ± 0.52 |
| **RobustZero** | **-83.48 ± 58.96** | **-87.92 ± 53.24** | **-84.68 ± 55.74** | **-9.57 ± 0.43** | **-12.16 ± 1.19** | **-11.43 ± 0.37** |
| Method | IEEE 8500-node | | | Racetrack | | |
| | Natural Reward | Worst-case Reward | Random-case Reward | Natural Reward | Worst-case Reward | Random-case Reward |
| RobustZero/cl | -1320 ± 30 | -1598 ± 36 | -1479 ± 25 | 422.60 ± 1.36 | 360.33 ± 1.35 | 378.65 ± 2.30 |
| RobustZero/$w1$ | -1200 ± 28 | -1488 ± 32 | -1369 ± 28 | 490.11 ± 1.32 | 400.00 ± 1.24 | 440.89 ± 2.31 |
| RobustZero/$w2$ | -1288 ± 29 | -1533 ± 34 | -1403 ± 26 | 448.70 ± 1.28 | 385.23 ± 1.34 | 414.55 ± 2.28 |
| RobustZero-$\mathcal{D}_{WC}$ | -1416 ± 27 | -1609 ± 35 | -1567 ± 25 | 380.26 ± 1.35 | 359.77 ± 1.11 | 368.75 ± 2.34 |
| RobustZero-$\mathcal{D}_{RC}$ | -1380 ± 28 | -1641 ± 37 | -1520 ± 24 | 392.30 ± 1.31 | 355.74 ± 1.20 | 370.02 ± 2.52 |
| **RobustZero** | **-1158 ± 30** | **-1374 ± 35** | **-1217 ± 27** | **520.60 ± 1.32** | **477.00 ± 1.26** | **494.00 ± 2.42** |

provided in Appendix C.4. Compared to RobustZero, we derive the following observations from Table 2:

- RobustZero/cl decreases the natural reward by 2.34%–25.71%, the worst-case reward by 3.67%–32.38%, and the random-case reward by 3.02%–30.47%. Thus, the self-supervised representation network clearly enhances the robustness of RobustZero, while ensuring a high natural reward.

- RobustZero/$w1$ decreases the natural reward by 0.61%–6.22%, the worst-case reward by 1.69%–19.25%, and the random-case reward by 0.60%–12.49%. This verifies that using a dynamic $w1$ to first enhance the defense capability to worst-case state perturbation and then to gradually adapt to random-case state perturbation is beneficial.

- RobustZero/$w2$ decreases the natural reward by 1.43%–20.90%, the worst-case reward by 2.63%–23.82%, and the random-case reward by 2.55%–19.17%. This is because an adaptive $w2$ enables the learned model to converge to flat minima with enhanced robustness to unseen states. A fixed $w2$ does not have this capability.

- RobustZero-$\mathcal{D}_{WC}$ decreases the natural reward by 7.04%–47.96%, the worst-case reward by 3.08%–32.58%, and the random-case reward by 7.32%–33.97%. Further, RobustZero-$\mathcal{D}_{RC}$ decreases the natural reward by 5.02%–42.11%, the worst-case reward by 4.94%–34.09%, and the random-case reward by 5.21%–33.51%. RobustZero-$\mathcal{D}_{WC}$ and RobustZero-$\mathcal{D}_{RC}$ both reduce RobustZero's performance notably. This is because the dynamic adjustment of $w1$ and $w2$ requires the interaction of two buffers. Using separate buffers makes $w1$ and $w2$ ineffective.

We further evaluate the performance of all variants of RobustZero against unforeseen perturbation strategies (see Appendix C.5). In addition, to assess the impacts of hyperparameter settings and attack radiuses on the performance of RobustZero, we conduct extensive experiments in Appendixes C.7 and C.8.

## 7. Conclusion and Future Work

We propose RobustZero, the first MuZero-class method designed to ensure robustness to state perturbations. It features a novel robust training framework and training method to defend against both worst-case and random-case state perturbations. The framework includes a self-supervised representation network that enables the generation of consistent policies before and after state perturbations. Further, a unique loss function enables robust training. In addition, an adaptive adjustment mechanism enables updates of RobustZero to offer a policy with high robustness to worst-case and random-case state perturbations. Extensive experiments on eight environments offer evidence that RobustZero advances the state-of-the-art methods at defending against worst-case and random-case state perturbations. In future research, it is of interest to develop a multi-agent RobustZero and apply it in Internet of Things scenarios.

## Acknowledgments

This work was supported in part by the Digital Energy Hub grant from the Association of Danish Industry and in part by Villum Investigator Grant S4OS under Grant 37819.

## Impact Statement

This paper advances the field of RL by enhancing the robustness of MuZero-class methods to state perturbations. To the best of our knowledge, this gives the first successful attempt in such a field. There are many potential societal consequences of our work, none which we feel must be specifically highlighted here.

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

## A. An Example in Energy System Subject to State Perturbations

The following example show why it is important to enhance the robustness of MuZero-class methods to state perturbations.
**Example 1.** We consider an environment encompassing a 6-bus power system, where the voltage (the state) of each bus must be in the range $[0.50, 1.00]$ p.u. (per unit). At time $t$, the real voltages of the six buses are $(0.95\ 0.55\ 0.78\ 0.86\ 0.87\ 1.00)$p.u. However, due to malicious attacks and noisy sensors, the agent receives perturbed voltage readings: $(0.96, 0.57, 0.80, 0.85, 0.89, 0.97)$ p.u. These perturbations affect the agent's actions, leading to different rewards and subsequent voltages. In particular, without perturbations, the reward is $0.89$ p.u. and the resulting voltages are $[0.94, 0.60, 0.75, 0.85, 0.85, 0.96]$ p.u. With perturbations, the reward drops to $0.22$ p.u., and the next set of voltages is $[0.90, 0.68, 0.77, 0.92, 0.80, 1.03]$ p.u., which violates the range constraint.

## B. Details of RobustZero

### B.1. Details of Parallel Data Collection

We consider parallel data collection to gain experience when states are suffering from state perturbations. Algorithm 1 provides the pseudocode for each round of data collection. Functions $WC(\cdot)$ and $RC(\cdot)$ define the data collection under worst-case and random-case state perturbations, respectively. Both Environment 1 and Environment 2 support multiple and parallel task execution to collect data. As a result, multiple $WC(\cdot)$ and $RC(\cdot)$ calls can be executed in parallel. For simplicity, we ignore episodes here. Additionally, $WC(\cdot)$ and $RC(\cdot)$ use the same NN parameters $\theta$.

We take function $WC(\cdot)$ as an example to illustrate data collection. Adversary 1 observes the real state $o_{\widetilde{t}}$ from Environment 1 and launches worst-case perturbation policy $\mathbb{A}_{\text{worst}}(o_{\widetilde{t}})$ to obtain the perturbed state $\widetilde{o}_{\widetilde{t}}$. Next, RobustZero agent receives $\widetilde{o}_{\widetilde{t}}$ that is further encoded as the initial hidden state (i.e., the root node) $\widetilde{s}_{\widetilde{t}}^0$ by the using the representation network

---

**Algorithm 1** Parallel Data Collection

**Input:** NN parameters $\theta$; number of sampled actions $\mathcal{K}$; end time of episode $\widetilde{T}$ and $\hat{T}$
**Output:** $\cup_{1 \le \widetilde{t} \le \widetilde{T}}(o_{\widetilde{t}}, \widetilde{o}_{\widetilde{t}}, \widetilde{\pi}_{\widetilde{t}}, \widetilde{a}_{\widetilde{t}+1}, \widetilde{\mu}_{\widetilde{t}+1}, o_{\widetilde{t}+1})$ and $\cup_{1 \le \hat{t} \le \hat{T}}(o_{\hat{t}}, \hat{o}_{\hat{t}}, \hat{\pi}_{\hat{t}}, \hat{a}_{\hat{t}+1}, \hat{\mu}_{\hat{t}+1}, o_{\hat{t}+1})$
**function** $WC(o_{\widetilde{t}}, \mathbb{A}_{\text{worst}}, \mathcal{H}_\theta, s\text{-}MCTS)$
  **for** $\widetilde{t} = 1$ **to** $\widetilde{T}$ **do**
    **Adversary 1 do:**
    observe $o_{\widetilde{t}}, \widetilde{o}_{\widetilde{t}} \leftarrow \mathbb{A}_{\text{worst}}(o_{\widetilde{t}})$
    **Agent of RobustZero do:**
    receive $\widetilde{o}_{\widetilde{t}}, \widetilde{s}_{\widetilde{t}}^0 \leftarrow \mathcal{H}_\theta(\widetilde{o}_{\widetilde{t}})$
    $\widetilde{\pi}_{\widetilde{t}} \leftarrow s\text{-}MCTS(\widetilde{s}_{\widetilde{t}}^0, \theta, \mathcal{K})$
    $\widetilde{a}_{\widetilde{t}+1} \leftarrow \widetilde{\pi}_{\widetilde{t}}$ `/* sample an action */`
    **Environment 1 do:**
    receive $\widetilde{a}_{\widetilde{t}+1}$, return $o_{\widetilde{t}+1}$ and $\widetilde{\mu}_{\widetilde{t}+1}$
  **end for**
  RETURN $\cup_{1 \le \widetilde{t} \le \widetilde{T}}(o_{\widetilde{t}}, \widetilde{o}_{\widetilde{t}}, \widetilde{\pi}_{\widetilde{t}}, \widetilde{a}_{\widetilde{t}+1}, \widetilde{\mu}_{\widetilde{t}+1}, o_{\widetilde{t}+1})$
**end function**
**function** $RC(o_{\hat{t}}, \mathbb{A}_{random}, \mathcal{H}_\theta, s\text{-}MCTS)$
  **for** $\hat{t} = 1$ **to** $\hat{T}$ **do**
    **Adversary 2 do:**
    observe $o_{\hat{t}}, \hat{o}_{\hat{t}} \leftarrow \mathbb{A}_{\text{random}}(o_{\hat{t}})$
    **Agent of RobustZero do:**
    receive $\hat{o}_{\hat{t}}, \hat{s}_{\hat{t}}^0 \leftarrow \mathcal{H}_\theta(\hat{o}_{\hat{t}})$
    $\hat{\pi}_{\hat{t}} \leftarrow s\text{-}MCTS(\hat{s}_{\hat{t}}^0, \theta, \mathcal{K})$
    $\hat{a}_{\hat{t}+1} \leftarrow \hat{\pi}_{\hat{t}}$
    **Environment 2 do:**
    receive $\hat{a}_{\hat{t}+1}$, generate $o_{\hat{t}+1}$ and $\hat{\mu}_{\hat{t}+1}$
  **end for**
  RETURN $\cup_{1 \le \hat{t} \le \hat{T}}(o_{\hat{t}}, \hat{o}_{\hat{t}}, \hat{\pi}_{\hat{t}}, \hat{a}_{\hat{t}+1}, \hat{\mu}_{\hat{t}+1}, o_{\hat{t}+1})$
**end function**

---

---

**Algorithm 2** Training process of RobustZero

> **Input:** Number of iterations $\mathcal{N}_{iter}$, number of update per iteration $N_u$, batch size $\mathcal{B}$, hyperparameters $\lambda_1$ and $\lambda_2$, and step-size $\varsigma$
> **Output:** $\theta$ and $\phi$
> Initialize NNs of $\theta$ and $\phi$, replay buffers $\mathcal{D}_{WC}$ and $\mathcal{D}_{RC}$, and $w1$ and $w2$ as 0
> **for** $i = 1$ **to** $\mathcal{N}_{iter}$ **do**
>    $\mathcal{D}_{WC} \leftarrow WC(\cdot), \mathcal{D}_{RC} \leftarrow RC(\cdot)$ `/* See WC(·) and RC(·) from Algorithm 1 */`
>    **for** $j = 1$ **to** $N_u$ **do**
>       $t' \leftarrow (i-1)N_u + j$
>       Sample a random mini-batch of $\mathcal{B}$ samples from $\mathcal{D}_{WC}$ and $\mathcal{D}_{RC}$, respectively
>       Compute $w1_{t'}$ according to Eq. 1 and Eq. 7
>       $w1_{t'} \leftarrow 0.1w1_{t'-1} + 0.9w1_{t'}$
>       $\mathcal{L}' \leftarrow \widetilde{\mathcal{L}} + w1_{t'} \cdot \hat{\mathcal{L}}$
>       $\nabla\widetilde{\mathcal{L}}_{\theta_{t'-1}}, \hat{\mathcal{L}}_{\theta_{t'-1}}, \nabla\widetilde{\mathcal{L}}_{\phi_{t'-1}}, \hat{\mathcal{L}}_{\phi_{t'-1}} \leftarrow backward(\mathcal{L}')$ `/* Calculate gradients */`
>       Compute $w2_{t'}$ according to Eq. 10
>       $w2_{t'} \leftarrow 0.1w2_{t'-1} + 0.9w2_{t'}$
>       $\theta_{t'} \leftarrow \theta_{t'-1} - \varsigma\big(\nabla\widetilde{\mathcal{L}}_{\theta_{t'-1}} + w1_{t'}\nabla\hat{\mathcal{L}}_{\theta_{t'-1}} + w2_{t'}\theta_{t'-1}\big)$
>       $\phi_{t'} \leftarrow \phi_{t'-1} - \varsigma\big(\nabla\widetilde{\mathcal{L}}_{\phi_{t'-1}} + w1_{t'}\nabla\hat{\mathcal{L}}_{\phi_{t'-1}}\big)$
>    **end for**
> **end for**
> RETURN $\theta$ and $\phi$

---

$\mathcal{H}_\theta(\widetilde{o_{\tilde{t}}})$. The sampled MCTS (Hubert et al., 2021) is used to obtain the search policy $\widetilde{\pi}_{\tilde{t}}$. The action $\widetilde{a}_{\tilde{t}+1}$ is determined by sampling an action from $\widetilde{\pi}_{\tilde{t}}$, with the same setting as in the literature (Hubert et al., 2021). Then, Environment 1 receives the action $\widetilde{a}_{\tilde{t}+1}$ and generates the immediate real reward $\widetilde{\mu}_{\tilde{t}+1}$ and the next state $o_{\tilde{t}+1}$. Finally, the data $\cup_{1 \le \tilde{t} \le \tilde{T}}(o_{\tilde{t}}, \widetilde{o}_{\tilde{t}}, \widetilde{\pi}_{\tilde{t}}, \widetilde{a}_{\tilde{t}+1}, \widetilde{\mu}_{\tilde{t}+1}, o_{\tilde{t}+1})$ is stored in the worst-case buffer $\mathcal{D}_{WC}$. A similar procedure is applied to function $RC(\cdot)$. The key differences are that the perturbed state is determined by using the random perturbation policy $\mathbb{A}_{\mathrm{random}}(o_{\hat{t}})$, and the data $\cup_{1 \le \hat{t} \le \hat{T}}(o_{\hat{t}}, \hat{o}_{\hat{t}}, \hat{\pi}_{\hat{t}}, \hat{a}_{\hat{t}+1}, \hat{\mu}_{\hat{t}+1}, o_{\hat{t}+1})$ are stored in the random-case buffer $\mathcal{D}_{RC}$.

### B.2. Pseudocode of the Training Process

The implementation of training the RobustZero is provided in Algorithm 2.

## C. Details of Experiments

### C.1. Details of Experimental Setup

We conduct experiments on an 8-core Intel Xeon E5-2640 v4 @ 2.40GHz CPU, with each node equipped with a GeForce RTX 3090 GPU, 2.40GHz processor, and 24GB RAM. Following existing studies (Zhang et al., 2021; Liu et al., 2024), we train 21 agents for each method with the same hyperparameters. The agent with median performance is reported for the purpose of reproducibility. Regrading attack radius, we set $\epsilon = 0.20$ for two classical control environments (i.e., CartPole and Pendulum), $\epsilon = 0.10$ for the three energy environments (i.e., IEEE 34-bus, IEEE 123-bus, and IEEE 8500-node), $\epsilon = 0.15$ for the three transportation environments (i.e., Highway, Racetrack, and Intersection), and $\epsilon = 0.075$ for Hopper, $\epsilon = 0.05$ for Walker2d, $\epsilon = 0.15$ for Halfcheetah, and $\epsilon = 0.15$ for Ant, respectively. Moreover, we use $l_\infty$ norm for the budget constraint.

We follow the original configurations for ATLA-PPO (Zhang et al., 2021) and PROTECTED (Liu et al., 2024). The configurations for S-MuZero, S-MuZero-worst, and S-MuZero-random are available on open science platforms alongside our project. The random perturbation is implemented by using Uniform noise that follows an uniform distribution $\mathcal{U}(-\epsilon, \epsilon)$.

### C.2. Major Comparison Studies on The Remaining Eight Environments

Following the results in Section 6.2, we further provide the corresponding results on the remaining eight environments. The average nature, worst-case, and random rewards for all methods on CartPole, IEEE 123-bus, Highway, Intersection, Hopper, Walker2d, HalfCheetah, and Ant are reported in Table 3. We observe that RobustZero consistently outperforms

*Table 3.* Main experimental results on CartPole, IEEE 123-bus, Highway, Intersection, Hopper, Walker2d, HalfCheetah, and Ant.

| Method | CartPole | | | IEEE 123-bus | | |
| --- | --- | --- | --- | --- | --- | --- |
| | Natural Reward | Worst-case Reward | Random-case Reward | Natural Reward | Worst-case Reward | Random-case Reward |
| ATLA-PPO | **500.00 ± 0.00** | 414.38 ± 12.13 | 420.26 ± 144.08 | -13.84 ± 0.06 | -15.24 ± 0.24 | -14.54 ± 0.04 |
| PROTECTED | **500.00 ± 0.00** | 430.22 ± 155.66 | 435.47 ± 162.66 | -12.29 ± 0.06 | -14.15 ± 0.27 | -13.42 ± 0.03 |
| S-MuZero | **500.00 ± 0.00** | 200.14 ± 18.01 | 359.98 ± 86.31 | **-9.77 ± 1.96** | -28.84 ± 5.84 | -19.63 ± 2.75 |
| S-MuZero-worst | 239.14 ± 6.24 | 428.64 ± 69.73 | 412.62 ± 16.42 | -16.67 ± 4.92 | -15.78 ± 3.99 | -18.61 ± 3.77 |
| S-MuZero-random | 444.14 ± 75.91 | 284.52 ± 78.70 | 448.24 ± 87.75 | -11.59 ± 2.88 | -23.02 ± 3.19 | -17.16 ± 4.12 |
| **RobustZero** | **500.00 ± 0.00** | **460.56 ± 56.33** | **490.06 ± 5.15** | -10.48 ± 1.19 | **-13.25 ± 1.43** | **-12.32 ± 2.00** |

| Method | Highway | | | Intersection | | |
| --- | --- | --- | --- | --- | --- | --- |
| | Natural Reward | Worst-case Reward | Random-case Reward | Natural Reward | Worst-case Reward | Random-case Reward |
| ATLA-PPO | 18.56 ± 0.50 | 18.68 ± 0.17 | 18.73 ± 0.08 | 2.32 ± 0.03 | 1.78 ± 0.29 | 1.87 ± 0.05 |
| PROTECTED | 20.26 ± 0.80 | 19.11 ± 0.14 | 19.48 ± 0.03 | 3.18 ± 0.07 | 2.05 ± 0.30 | 2.30 ± 0.10 |
| S-MuZero | **25.83 ± 3.16** | 15.91 ± 4.93 | 17.47 ± 2.94 | **3.63 ± 0.30** | 1.56 ± 0.32 | 1.68 ± 0.03 |
| S-MuZero-worst | 13.64 ± 5.36 | 18.62 ± 3.59 | 17.89 ± 1.95 | 1.52 ± 0.18 | 1.98 ± 0.50 | 1.84 ± 0.80 |
| S-MuZero-random | 14.47 ± 3.64 | 17.49 ± 4.64 | 18.19 ± 2.36 | 2.07 ± 0.68 | 1.92 ± 0.13 | 2.13 ± 0.21 |
| **RobustZero** | 23.24 ± 1.32 | **20.95 ± 1.26** | **22.09 ± 1.42** | 3.43 ± 0.26 | **2.87 ± 1.48** | **3.15 ± 0.34** |

| Method | Hopper | | | Walker2d | | |
| --- | --- | --- | --- | --- | --- | --- |
| | Natural Reward | Worst-case Reward | Random-case Reward | Natural Reward | Worst-case Reward | Random-case Reward |
| ATLA-PPO | 3311 ± 310 | 1736 ± 360 | 3167 ± 180 | 3831 ± 200 | 3659 ± 210 | 3898 ± 300 |
| PROTECTED | 3629 ± 300 | 2533 ± 251 | 3538 ± 310 | 6311 ± 112 | 6010 ± 321 | 6142 ± 150 |
| S-MuZero | **3672 ± 316** | 1452 ± 390 | 2454 ± 284 | **6430 ± 280** | 890 ± 180 | 2615 ± 200 |
| S-MuZero-worst | 1675 ± 332 | 1810 ± 194 | 1756 ± 132 | 1315 ± 120 | 1508 ± 110 | 1435 ± 130 |
| S-MuZero-random | 2678 ± 264 | 1732 ± 264 | 2589 ± 306 | 2627 ± 180 | 1158 ± 200 | 2640 ± 120 |
| **RobustZero** | 3660 ± 132 | **2577 ± 126** | **3553 ± 242** | 6384 ± 230 | **6096 ± 350** | **6322 ± 320** |

| Method | HalfCheetah | | | Ant | | |
| --- | --- | --- | --- | --- | --- | --- |
| | Natural Reward | Worst-case Reward | Random-case Reward | Natural Reward | Worst-case Reward | Random-case Reward |
| ATLA-PPO | 6106 ± 210 | 5114 ± 230 | 6112 ± 150 | 5335 ± 150 | 3780 ± 120 | 5352 ± 270 |
| PROTECTED | 7044 ± 168 | 5186 ± 175 | 6258 ± 209 | 5736 ± 280 | 4582 ± 190 | 5586 ± 190 |
| S-MuZero | **7185 ± 360** | 2524 ± 272 | 3245 ± 184 | **5830 ± 200** | 510 ± 250 | 3730 ± 150 |
| S-MuZero-worst | 2714 ± 302 | 2830 ± 149 | 2786 ± 129 | 1206 ± 260 | 1423 ± 210 | 1330 ± 100 |
| S-MuZero-random | 3486 ± 226 | 2692 ± 246 | 3575 ± 286 | 3894 ± 200 | 805 ± 180 | 3811 ± 150 |
| **RobustZero** | 7100 ± 232 | **5395 ± 162** | **6394 ± 192** | 5782 ± 130 | **4686 ± 250** | **5692 ± 220** |

all baselines at defending against state perturbations across all environments. Specially, RobustZero and S-Muzero still obtain similar nature rewards, where both of them are higher than those obtained by using other baselines. The exception is on CartPole, where four methods are able to obtain the optimal solution. Moreover, although S-MuZero achieves the best natural rewards, it obtains low worst-case and random-case rewards. This implies that s-MuZero is very sensitive to state perturbations due to the lack of a defense strategy. This is the motivation for developing RobustZero. As expected, RobustZero significantly enhances the robustness to state perturbations with highest worst-case and random-case rewards compared to all baselines, while maintaining comparable natural rewards compared to S-MuZero. As observed in Table 3, RobustZero increases: 1) the worst-case reward by 7.05%–130.12% on CartPole, by 6.79%–117.66% on IEEE 123-bus, by 9.63%–31.68% on Highway, by 40.00%–83.97% on Intersection, by 1.7%–77.5% on Hopper, by 1.4%–584.9% on Walker2d, by 4.0%–113.7% on HalfCheetah, and by 2.3%–818.8% on Ant; and 2) random-case reward by 12.54%–36.14% on CartPole, by 8.93%–59.33% on IEEE 123-bus, by 13.40%–26.45% on Highway, by 36.96%–87.50% on Intersection, and by 0.4%–102.3% on Hopper, by 2.9%–340.6% on Walker2d, by 2.2%–129.5% on HalfCheetah, and by 1.9%–328.0% on Ant.

### C.3. Selection of $w2$ in Ablation Study

We set deterministic values for $w2$ in different environments based on the average of the corresponding dynamic $w2$, selecting the best from five points near this average. The results of natural, worst-case, and random-case rewards by using different values of $w2$ on the eight environments are provided in Fig. 2. As observed, RobustZero/$w2$ obtains the best natural, worst-case, and random-case rewards when it is set to 5e-6, 5e-6, 5e-8, 5e-8, 5e-8, 1e-5, 1e-5, 1e-5, 5e-6, 5e-6, 1e-6, and 1e-6 for CartPole, Pendulum, IEEE 34-bus, IEEE 123-bus, IEEE 8500-node, Highway, Intersection, Racetrack, and Hopper, Walker2d, HalfCheetah, and Ant, respectively.

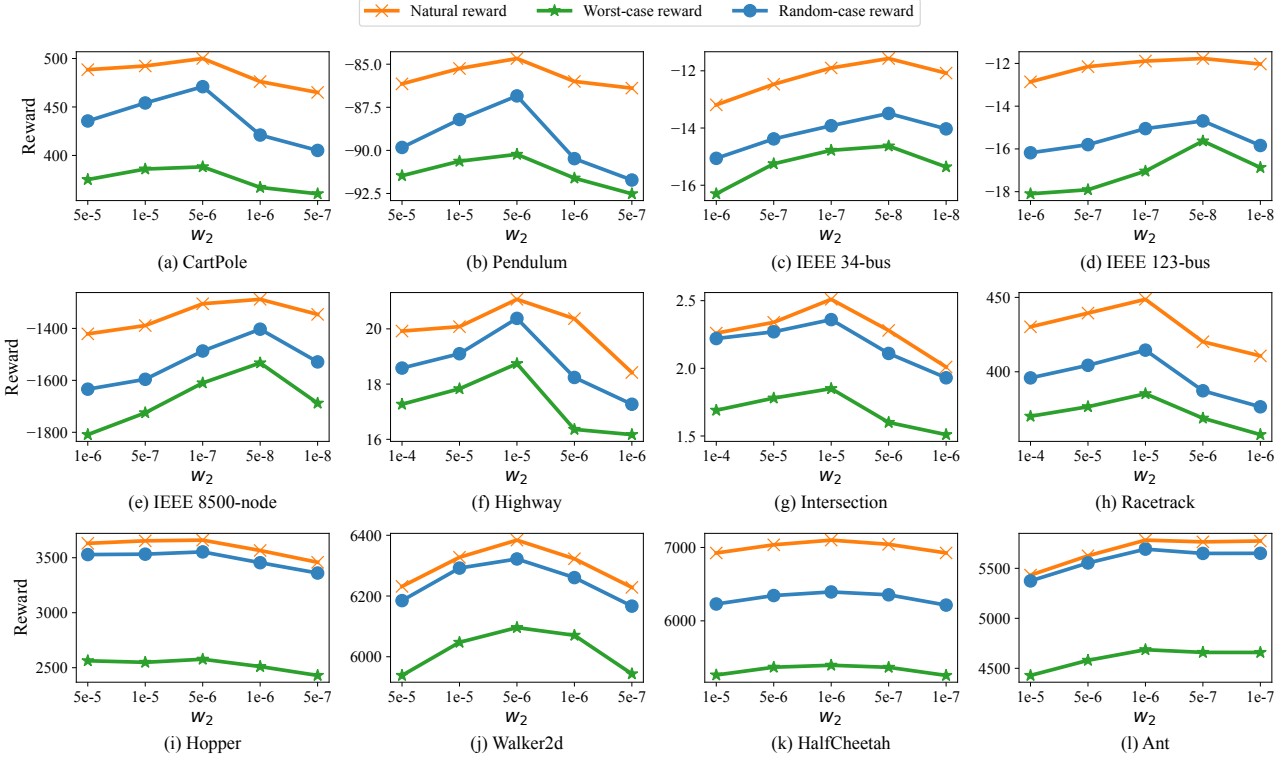

*Figure 2.* The selection of $w2$.

### C.4. Major Ablation Study on The Remaining Eight Environments

Following the results in Section 6.3, we further provide the corresponding ablation study on the remaining eight environments. Specially, the average nature, worst-case, and random rewards for RobustZero/cl, RobustZero/$w1$, RobustZero/$w2$, RobustZero-$\mathcal{D}_{WC}$, and RobustZero-$\mathcal{D}_{RC}$ on CartPole, IEEE 123-bus, Highway, Intersection, Hopper, Walker2d, HalfCheetah, and Ant are reported in Table 4. Regrading RobustZero/$w1$, we maintain $w1 = 1$. Regarding RobustZero/$w2$, the settings of $w2$ is provided in Appendix C.3. Of note, RobustZero/cl, RobustZero/$w1$, and RobustZero/$w2$ can obtain the optimal natural reward on CartPole. To better analyze the effects of the self-supervised representation network and the adaptive adjustment mechanism, we do not account for the corresponding natural rewards on CartPole in following statistics. Next, compared to RobustZero, we have the following statistic results: 1) RobustZero/cl decreases the natural reward by 2.65%–61.79%, the worst-case reward by 4.0%–61.24%, and the random-case reward by 1.2%–51.44%; 2) RobustZero/w1 decreases the natural reward by 0.7%–18.28%, the worst-case reward by 1.2%–42.79%, and the random-case reward by 0.8%–13.6%; 3) RobustZero/w2 decreases the natural reward by 1.9%–36.65%, the worst-case reward by 1.5%–55.14%, and the random-case reward by 2.6%–33.47%; 4) RobustZero-$\mathcal{D}_{WC}$ decreases the natural reward by 7.8%–67.32%, the worst-case reward by 3.0%–68.82%, and the random-case reward by 9.6%–61.54%; and 5) RobustZero-$\mathcal{D}_{RC}$ decreases the natural reward by 3.6%–49.13%, the worst-case reward by 4.7%–73.94%, and the random-case reward by 6.3%–57.5%. These results are consistent to those provided in Section 6.3. This further verifies that the absence of any of the self-supervised representation network, the adaptive adjustment mechanism, and the parallel use of worst-case and random-case buffers will diminish the robustness of RobustZero.

### C.5. Performance against Unforeseen Perturbation Policies

In order to assess the generalization capability to unforeseen perturbations policies, we evaluate the performance of RobustZero, all baselines, and all variants of RobustZero to handle two additional perturbation policies (including BA-DDPG (Russo & Proutiere, 2021) and random perturbation based on Gaussian noise (RP-G)) that are not included during training. BA-DDPG is a kind of worst-case perturbation policy, while RP-G is a kind of random-case perturbation policy that follows a Gaussian distribution $\mathcal{N}(0, \epsilon/3)$. We adhere to the original neural network architectures and parameters to implement BA-DDPG across the eight environments. The results are listed in Table 5 and Table 6. We have the following

*Table 4.* Ablation study on CartPole, IEEE 123-bus, Highway, Intersection, Hopper, Walker2d, HalfCheetah, and Ant.

| Method | CartPole | | | IEEE 123-bus | | |
|---|---|---|---|---|---|---|
| | Natural Reward | Worst-case Reward | Random-case Reward | Natural Reward | Worst-case Reward | Random-case Reward |
| RobustZero/cl | **500.00 ± 0.00** | 374.15 ± 12.04 | 460.24 ± 100.12 | -12.12 ± 1.00 | -15.96 ± 1.16 | -15.00 ± 2.12 |
| RobustZero/$w1$ | **500.00 ± 0.00** | 396.30 ± 50.42 | 480.12 ± 80.56 | -11.04 ± 1.21 | -14.11 ± 1.20 | -13.38 ± 2.06 |
| RobustZero/$w2$ | **500.00 ± 0.00** | 388.23 ± 30.42 | 470.94 ± 28.46 | -11.77 ± 1.06 | -15.63 ± 1.18 | -14.69 ± 2.10 |
| RobustZero-$\mathcal{D}_{WC}$ | 452.00 ± 7.00 | 369.70 ± 76.00 | 412.00 ± 20.00 | -14.01 ± 1.02 | -16.48 ± 2.10 | -15.85 ± 2.00 |
| RobustZero-$\mathcal{D}_{RC}$ | 463.70 ± 60.00 | 360.56 ± 62.12 | 435.70 ± 80.12 | -13.87 ± 1.01 | -16.53 ± 1.30 | -15.49 ± 1.92 |
| **RobustZero** | **500.00 ± 0.00** | **460.56 ± 56.33** | **490.06 ± 5.15** | **-10.48 ± 1.19** | **-13.25 ± 1.43** | **-12.32 ± 2.00** |

| Method | Highway | | | Intersection | | |
|---|---|---|---|---|---|---|
| | Natural Reward | Worst-case Reward | Random-case Reward | Natural Reward | Worst-case Reward | Random-case Reward |
| RobustZero/cl | 20.03 ± 1.40 | 17.97 ± 1.25 | 19.23 ± 2.09 | 2.12 ± 0.36 | 1.78 ± 1.45 | 2.08 ± 0.30 |
| RobustZero/$w1$ | 22.12 ± 1.42 | 19.42 ± 1.30 | 21.05 ± 2.07 | 2.90 ± 0.32 | 2.01 ± 1.40 | 2.81 ± 0.31 |
| RobustZero/$w2$ | 21.07 ± 1.36 | 18.69 ± 1.25 | 20.38 ± 2.10 | 2.51 ± 0.28 | 1.85 ± 1.34 | 2.36 ± 0.33 |
| RobustZero-$\mathcal{D}_{WC}$ | 18.92 ± 1.88 | 18.84 ± 2.39 | 18.13 ± 1.91 | 2.05 ± 0.35 | 1.70 ± 1.11 | 1.95 ± 0.34 |
| RobustZero-$\mathcal{D}_{RC}$ | 19.36 ± 1.31 | 18.31 ± 1.09 | 18.90 ± 1.52 | 2.30 ± 0.31 | 1.65 ± 1.20 | 2.00 ± 0.52 |
| **RobustZero** | **23.24 ± 1.32** | **20.95 ± 1.26** | **22.09 ± 1.42** | **3.43 ± 0.26** | **2.87 ± 1.48** | **3.15 ± 0.34** |

| Method | Hopper | | | Walker2d | | |
|---|---|---|---|---|---|---|
| | Natural Reward | Worst-case Reward | Random-case Reward | Natural Reward | Worst-case Reward | Random-case Reward |
| **RobustZero**/cl | 3561 ± 133 | 2478 ± 137 | 3511 ± 227 | 5377 ± 162 | 4998 ± 328 | 5007 ± 295 |
| **RobustZero**/$w1$ | 3634 ± 128 | 2546 ± 119 | 3524 ± 227 | 5986 ± 224 | 5592 ± 379 | 5829 ± 232 |
| **RobustZero**/$w2$ | 3592 ± 126 | 2538 ± 133 | 3463 ± 241 | 5157 ± 178 | 4991 ± 304 | 5252 ± 214 |
| **RobustZero**-$\mathcal{D}_{WC}$ | 3396 ± 122 | 2501 ± 123 | 3241 ± 234 | 4297 ± 244 | 4738 ± 214 | 5046 ± 343 |
| **RobustZero**-$\mathcal{D}_{RC}$ | 3534 ± 121 | 2462 ± 140 | 3343 ± 269 | 4436 ± 345 | 4886 ± 317 | 4891 ± 433 |
| **RobustZero** | **3660 ± 132** | **2577 ± 126** | **3553± 242** | **6384 ± 230** | **6096 ± 350** | **6322 ± 320** |

| Method | HalfCheetah | | | Ant | | |
|---|---|---|---|---|---|---|
| | Natural Reward | Worst-case Reward | Random-case Reward | Natural Reward | Worst-case Reward | Random-case Reward |
| **RobustZero**/cl | 6080 ± 219 | 4746 ± 164 | 5165 ± 173 | 4743 ± 144 | 3852 ± 251 | 4513 ± 204 |
| **RobustZero**/$w1$ | 6891 ± 209 | 5006 ± 138 | 5628 ± 198 | 5467 ± 131 | 3945 ± 256 | 4992 ± 198 |
| **RobustZero**/$w2$ | 6486 ± 224 | 4620 ± 150 | 5510 ± 187 | 4959 ± 116 | 3789 ± 271 | 4938 ± 202 |
| **RobustZero**-$\mathcal{D}_{WC}$ | 5636 ± 217 | 4551 ± 158 | 4815 ± 178 | 4803 ± 130 | 3699 ± 215 | 4392 ± 221 |
| **RobustZero**-$\mathcal{D}_{RC}$ | 6201 ± 230 | 4507 ± 167 | 5431 ± 177 | 4351 ± 126 | 3580 ± 249 | 4411 ± 214 |
| **RobustZero** | **7100 ± 232** | **5395 ± 162** | **6394 ± 192** | **5782 ± 130** | **4686 ± 250** | **5692 ± 220** |

observations: 1) RobustZero achieves a higher worst-case reward than all baselines and RobustZero variants across all environments. Specially, RobustZero increases the worst-case reward by 7.8%–119.85% on CartPole, by 2.32%–76.31% on Pendulum, by 2.78%–101.01% on IEEE 34-bus, by 6.33%–99.31% on IEEE 123-bus, by 2.04%–34.06% on IEEE 8500, by 3.33%–29.35% on Highway, by 36.65%–86.42% on Intersection, by 11.09%–32.79% on Racetrack, by 2.5%–80.0% on Hopper, by 1.3%–583.3% on Walker2d, by 3.9%–114.0% on HalfCheetah, and by 2.2%–817.5% on Ant; and 2) RobustZero achieves a higher random-case reward than all baselines and RobustZero variants across all environments, increasing the random-case reward by 1.64%–33.78% on CartPole, by 0.88%–100.88% on Pendulum, by 6.31%–67.02% on IEEE 34-bus, by 7.17%–57.00% on IEEE 123-bus, by 3.9%–32.92% on IEEE 8500, by 4.39%–23.49% on Highway, by 10.88%–88.44% on Intersection, by 8.93%–35.69% on Racetrack, by 2.5%–80.0% on Hopper, by 1.3%–583.3% on Walker2d, by 3.9%–114.0% on HalfCheetah, and by 2.2%–817.5% on Ant. Thus, RobustZero still outperforms all baselines and RobustZero variants to unforeseen perturbations policies.

## C.6. Time and Sampling Efficiency Analysis

We measure the average training time per iteration (TT), sampling time per step (ST), testing time per step (TeT), sampling time per episode (STE), and the number of samples per episode (NSE). The results are provided in Table 7. We report the relationship between the number of environment samples and the natural, worst-case, and random-case rewards for RobustZero and the two robust model-free baselines: ATLA-PPO and PROTECTED. The results are presented in Table 8 and Table 9. Of note, the number of samples per episode in RobustZero differs from those used in the two baselines. Therefore, while the numbers of samples reported in Table 8 and Table 9 are similar across methods, they are not exactly the same. As observed from Table 7, ATLA-PPO and PROTECTED exhibit similar TT, ST, TeT, and STE across all

*Table 5.* Performance comparison of different methods against unforeseen perturbation policies on CartPole, Pendulum, IEEE 34-bus, IEEE 123-bus, IEEE 8500-node, Highway, Intersection, and Racetrack.

| Method | CartPole | | Pendulum | |
| --- | --- | --- | --- | --- |
| | Worst-case Reward (BA-DDPG) | Random-case Reward (RP-G) | Worst-case Reward (BA-DDPG) | Random-case Reward (RP-G) |
| ATLA-PPO | $430.26 \pm 12.78$ | $429.00 \pm 100.00$ | $-93.96 \pm 61.49$ | $-91.00 \pm 70.00$ |
| PROTECTED | $441.39 \pm 72.94$ | $440.10 \pm 90.00$ | $-90.38 \pm 32.28$ | $-90.10 \pm 55.00$ |
| S-MuZero | $216.43 \pm 17.32$ | $370.00 \pm 66.00$ | $-150.32 \pm 89.46$ | $-161.38 \pm 55.00$ |
| S-MuZero-worst | $431.26 \pm 66.89$ | $417.11 \pm 20.00$ | $-88.48 \pm 42.48$ | $-168.24 \pm 62.00$ |
| S-MuZero-random | $322.64 \pm 38.49$ | $460.78 \pm 90.00$ | $-134.79 \pm 48.51$ | $-110.98 \pm 98.00$ |
| RobustZero/cl | $385.76 \pm 9.42$ | $472.12 \pm 90.16$ | $-89.61 \pm 59.49$ | $-86.12 \pm 58.89$ |
| RobustZero/$w1$ | $411.43 \pm 44.87$ | $487.00 \pm 88.18$ | $-87.24 \pm 58.11$ | $-84.49 \pm 53.62$ |
| RobustZero/$w2$ | $405.39 \pm 29.37$ | $475.30 \pm 30.00$ | $-88.49 \pm 46.32$ | $-85.29 \pm 57.68$ |
| RobustZero-$\mathcal{D}_{WC}$ | $383.24 \pm 63.98$ | $419.48 \pm 19.32$ | $-90.12 \pm 51.36$ | $-89.90 \pm 56.00$ |
| RobustZero-$\mathcal{D}_{RC}$ | $379.42 \pm 45.94$ | $451.30 \pm 80.00$ | $-91.59 \pm 58.29$ | $-88.06 \pm 55.75$ |
| **RobustZero** | $\mathbf{475.82 \pm 39.28}$ | $\mathbf{495.00 \pm 6.00}$ | $\mathbf{-85.26 \pm 36.24}$ | $\mathbf{-83.75 \pm 55.15}$ |

| Method | IEEE 34-bus | | IEEE 123-bus | |
| --- | --- | --- | --- | --- |
| | Worst-case Reward (BA-DDPG) | Random-case Reward (RP-G) | Worst-case Reward (BA-DDPG) | Random-case Reward (RP-G) |
| ATLA-PPO | $-13.23 \pm 0.14$ | $-13.15 \pm 0.04$ | $-14.02 \pm 0.10$ | $-14.31 \pm 0.01$ |
| PROTECTED | $-12.19 \pm 0.25$ | $-12.20 \pm 0.12$ | $-13.94 \pm 0.13$ | $-13.26 \pm 0.04$ |
| S-MuZero | $-23.84 \pm 4.53$ | $-18.79 \pm 2.00$ | $-26.13 \pm 4.93$ | $-19.28 \pm 3.42$ |
| S-MuZero-worst | $-14.92 \pm 0.33$ | $-15.39 \pm 1.90$ | $-15.66 \pm 2.60$ | $-18.51 \pm 2.61$ |
| S-MuZero-random | $-15.16 \pm 0.83$ | $-14.12 \pm 0.30$ | $-21.43 \pm 2.18$ | $-16.62 \pm 3.49$ |
| RobustZero/cl | $-14.84 \pm 0.92$ | $-13.81 \pm 0.59$ | $-15.86 \pm 0.95$ | $-14.89 \pm 2.39$ |
| RobustZero/$w1$ | $-13.14 \pm 1.11$ | $-11.96 \pm 0.45$ | $-13.96 \pm 1.95$ | $-13.16 \pm 2.40$ |
| RobustZero/$w2$ | $-14.02 \pm 0.73$ | $-13.23 \pm 0.68$ | $-15.33 \pm 1.68$ | $-14.23 \pm 2.38$ |
| RobustZero-$\mathcal{D}_{WC}$ | $-14.43 \pm 0.96$ | $-14.56 \pm 0.51$ | $-15.87 \pm 1.07$ | $-15.68 \pm 2.51$ |
| RobustZero-$\mathcal{D}_{RC}$ | $-14.75 \pm 0.87$ | $-14.03 \pm 0.60$ | $-16.24 \pm 1.78$ | $-15.20 \pm 2.50$ |
| **RobustZero** | $\mathbf{-11.86 \pm 0.81}$ | $\mathbf{-11.25 \pm 0.30}$ | $\mathbf{-13.11 \pm 1.74}$ | $\mathbf{-12.28 \pm 2.96}$ |

| Method | IEEE 8500-node | | Highway | |
| --- | --- | --- | --- | --- |
| | Worst-case Reward (BA-DDPG) | Random-case Reward (RP-G) | Worst-case Reward (BA-DDPG) | Random-case Reward (RP-G) |
| ATLA-PPO | $-1392 \pm 11$ | $-1288 \pm 7$ | $18.79 \pm 0.21$ | $18.87 \pm 0.05$ |
| PROTECTED | $-1351 \pm 14$ | $-1253 \pm 6$ | $19.48 \pm 0.18$ | $19.94 \pm 0.06$ |
| S-MuZero | $-1775 \pm 23$ | $-1603 \pm 15$ | $16.56 \pm 3.98$ | $17.92 \pm 2.80$ |
| S-MuZero-worst | $-1452 \pm 15$ | $-1507 \pm 9$ | $18.68 \pm 2.73$ | $18.02 \pm 1.72$ |
| S-MuZero-random | $-1473 \pm 12$ | $-1314 \pm 5$ | $17.94 \pm 4.45$ | $18.64 \pm 2.00$ |
| RobustZero/cl | $-1502 \pm 21$ | $-1434 \pm 27$ | $18.83 \pm 1.02$ | $19.62 \pm 1.32$ |
| RobustZero/$w1$ | $-1445 \pm 38$ | $-1334 \pm 26$ | $20.73 \pm 1.56$ | $21.20 \pm 1.36$ |
| RobustZero/$w2$ | $-1485 \pm 29$ | $-1389 \pm 25$ | $18.74 \pm 0.92$ | $20.59 \pm 1.24$ |
| RobustZero-$\mathcal{D}_{WC}$ | $-1556 \pm 41$ | $-1551 \pm 28$ | $18.96 \pm 1.85$ | $18.53 \pm 1.02$ |
| RobustZero-$\mathcal{D}_{RC}$ | $-1612 \pm 37$ | $-1492 \pm 26$ | $18.54 \pm 1.43$ | $19.08 \pm 1.60$ |
| **RobustZero** | $\mathbf{-1324 \pm 26}$ | $\mathbf{-1206 \pm 27}$ | $\mathbf{21.42 \pm 1.35}$ | $\mathbf{22.13 \pm 2.10}$ |

| Method | Intersection | | Racetrack | |
| --- | --- | --- | --- | --- |
| | Worst-case Reward (BA-DDPG) | Random-case Reward (RP-G) | Worst-case Reward (BA-DDPG) | Random-case Reward (RP-G) |
| ATLA-PPO | $1.81 \pm 0.36$ | $1.95 \pm 0.06$ | $392.62 \pm 0.11$ | $400.11 \pm 0.03$ |
| PROTECTED | $2.11 \pm 0.31$ | $2.36 \pm 0.14$ | $433.17 \pm 4.71$ | $463.95 \pm 6.00$ |
| S-MuZero | $1.62 \pm 0.32$ | $1.73 \pm 0.12$ | $365.82 \pm 6.54$ | $376.12 \pm 4.00$ |
| S-MuZero-worst | $2.04 \pm 0.58$ | $1.92 \pm 0.44$ | $418.88 \pm 5.94$ | $401.49 \pm 6.38$ |
| S-MuZero-random | $2.06 \pm 0.14$ | $2.26 \pm 0.32$ | $403.42 \pm 9.16$ | $426.10 \pm 6.00$ |
| RobustZero/cl | $1.88 \pm 1.32$ | $2.13 \pm 0.36$ | $368.84 \pm 1.02$ | $389.57 \pm 2.12$ |
| RobustZero/$w1$ | $2.21 \pm 1.96$ | $2.94 \pm 0.49$ | $421.38 \pm 1.15$ | $450.75 \pm 2.06$ |
| RobustZero/$w2$ | $2.04 \pm 1.32$ | $2.46 \pm 0.27$ | $393.46 \pm 1.03$ | $429.30 \pm 2.14$ |
| RobustZero-$\mathcal{D}_{WC}$ | $1.93 \pm 1.36$ | $2.11 \pm 0.31$ | $371.96 \pm 1.54$ | $372.48 \pm 1.65$ |
| RobustZero-$\mathcal{D}_{RC}$ | $1.84 \pm 1.26$ | $2.18 \pm 0.60$ | $362.39 \pm 1.29$ | $381.90 \pm 2.00$ |
| **RobustZero** | $\mathbf{3.02 \pm 0.95}$ | $\mathbf{3.26 \pm 0.38}$ | $\mathbf{481.20 \pm 1.00}$ | $\mathbf{505.40 \pm 2.00}$ |

environments, while S-MuZero, S-MuZero-worst, S-MuZero-random, and RobustZero show similar TT, ST, TeT, and STE within their group. ATLA-PPO and PROTECTED are model-free DRL methods, which generally suffer from low sampling efficiency, as indicated by their significantly higher NSE, meaning they require a larger number of interactions

*Table 6.* Performance comparison of different methods against unforeseen perturbation policies on Hopper, Walker2d, HalfCheetah, and Ant.

| Method | Hopper | | Walker2d | |
| --- | --- | --- | --- | --- |
| | Worst-case Reward (BA-DDPG) | Random-case Reward (RP-G) | Worst-case Reward (BA-DDPG) | Random-case Reward (RP-G) |
| ATLA-PPO | $1744 \pm 44$ | $3187 \pm 50$ | $3695 \pm 746$ | $3928 \pm 276$ |
| PROTECTED | $2546 \pm 51$ | $3567 \pm 43$ | $6055 \pm 632$ | $6176 \pm 328$ |
| S-MuZero | $1462 \pm 93$ | $2473 \pm 105$ | $898 \pm 831$ | $2635 \pm 367$ |
| S-MuZero-worst | $1826 \pm 60$ | $1770 \pm 64$ | $1517 \pm 549$ | $1444 \pm 249$ |
| S-MuZero-random | $1742 \pm 50$ | $2607 \pm 31$ | $1169 \pm 912$ | $2658 \pm 299$ |
| **RobustZero**/cl | $2497 \pm 77$ | $3541 \pm 181$ | $5039 \pm 193$ | $5051 \pm 188$ |
| **RobustZero**/$w1$ | $2568 \pm 153$ | $3543 \pm 180$ | $5633 \pm 318$ | $5880 \pm 176$ |
| **RobustZero**/$w2$ | $2553 \pm 117$ | $3457 \pm 169$ | $5034 \pm 171$ | $5304 \pm 187$ |
| **RobustZero-**$\mathcal{D}_{WC}$ | $2524 \pm 167$ | $3267 \pm 209$ | $4768 \pm 394$ | $5090 \pm 145$ |
| **RobustZero-**$\mathcal{D}_{RC}$ | $2478 \pm 144$ | $3374 \pm 204$ | $4926 \pm 265$ | $4937 \pm 238$ |
| **RobustZero** | $\mathbf{2632 \pm 104}$ | $\mathbf{3593 \pm 192}$ | $\mathbf{6136 \pm 264}$ | $\mathbf{6369 \pm 299}$ |
| Method | HalfCheetah | | Ant | |
| | Worst-case Reward (BA-DDPG) | Random-case Reward (RP-G) | Worst-case Reward (BA-DDPG) | Random-case Reward (RP-G) |
| ATLA-PPO | $5145 \pm 256$ | $6170 \pm 227$ | $3815 \pm 125$ | $5384 \pm 603$ |
| PROTECTED | $5229 \pm 251$ | $6293 \pm 269$ | $4616 \pm 109$ | $5630 \pm 618$ |
| S-MuZero | $2538 \pm 257$ | $3266 \pm 257$ | $514 \pm 150$ | $3760 \pm 431$ |
| S-MuZero-worst | $2844 \pm 297$ | $2803 \pm 226$ | $1436 \pm 286$ | $1341 \pm 699$ |
| S-MuZero-random | $2706 \pm 222$ | $3599 \pm 144$ | $812 \pm 217$ | $3846 \pm 672$ |
| **RobustZero**/cl | $4788 \pm 221$ | $5200 \pm 186$ | $3884 \pm 215$ | $4547 \pm 248$ |
| **RobustZero**/$w1$ | $5043 \pm 329$ | $5679 \pm 236$ | $3983 \pm 271$ | $5034 \pm 239$ |
| **RobustZero**/$w2$ | $4651 \pm 206$ | $5550 \pm 147$ | $3821 \pm 240$ | $4973 \pm 206$ |
| **RobustZero-**$\mathcal{D}_{WC}$ | $4583 \pm 228$ | $4852 \pm 160$ | $3731 \pm 380$ | $4433 \pm 156$ |
| **RobustZero-**$\mathcal{D}_{RC}$ | $4537 \pm 202$ | $5464 \pm 297$ | $3608 \pm 288$ | $4448 \pm 233$ |
| **RobustZero** | $\mathbf{5432 \pm 154}$ | $\mathbf{6404 \pm 189}$ | $\mathbf{4716 \pm 232}$ | $\mathbf{5712 \pm 219}$ |

with environments. In contrast, RobustZero is model-based DRL methods, characterized by higher sampling efficiency. From Table 8 and Table 9, by using similar samples, RobustZero achieves higher rewards compared to ATLA-PPO and PROTECTED, which further demonstrates its superior sample efficiency. The higher sample efficiency of RobustZero makes it more suitable for real-world applications. Further, MuZero-class methods trade time efficiency for accuracy of decision-making (high rewards), by leveraging the learned models and MCTS to plan ahead. This explains why TT, ST, TeT, and STE for ATLA-PPO and PROTECTED are lower than those for S-MuZero, S-MuZero-worst, S-MuZero-random, and RobustZero. However, it is important to note that TT and STE for all methods remain within a comparable range, indicating that the time efficiency of S-MuZero, S-MuZero-worst, S-MuZero-random, and RobustZero does not significantly diminish when compared with ATLA-PPO and PROTECTED. On the contrary, S-MuZero, S-MuZero-worst, S-MuZero-random, and RobustZero methods offer greatly improved sampling efficiency, requiring fewer samples compared to ATLA-PPO and PROTECTED. Additionally, TT, ST, TeT, and STE of RobustZero are slightly higher than those of S-MuZero, S-MuZero-worst, and S-MuZero-random, due to the integration of self-supervised representation learning and an adaptive adjustment mechanism, which requires additional computational resources. Nevertheless, RobustZero still maintains comparable TT, ST, TeT, and STE, while demonstrating much stronger robustness to worst-case and random-case perturbations.

## C.7. Hyperparameter Analysis

We study how the two hyperparameters $\lambda_1$ and $\lambda_2$ affect the adaptive adjustment process.

- **Impact of $\lambda_1$.** Fig. 3 presents the average natural, worst-case, and random-case rewards of RobustZero with $\lambda_1$ ranging from 0.25 to 1.5. For sake of distinction, we reuse B1-worst-case reward and B2-worst-case reward to represent the worst-case rewards obtained under the perturbation policies by using ATLA-PPO and BA-DDPG, respectively. Similarly, we reuse U-random-case reward and G-random-case reward to represent the random-case rewards obtained under the perturbation policies by using Uniform noise and RP-G, respectively. We observe that the natural reward shows irregular changes, the worst-case reward increases as $\lambda_1$ increases, and the random-case reward decreases as $\lambda_1$ increases. This is because $\lambda_1$ controls the rate of change of $w1_{t'}$ (see Eq. 7), which does not impact the natural reward directly. Moreover, for the same $\widetilde{\mathcal{L}}_{ctr}$, an increase in $\lambda_1$ reduces the weight of the random-case loss term, i.e., $w1_{t'}$. This, in turn, decreases the ability to defend against such state perturbations. Thus, the random-case reward decreases,

*Table 7.* Performance metrics across all environments.

| Method | CartPole | | | | | Pendulum | | | | |
|---|---|---|---|---|---|---|---|---|---|---|
| | TT | ST | TeT | STE | NSE | TT | ST | TeT | STE | NSE |
| ATLA-PPO | 0.4051 | 0.0036 | 0.0034 | 7.3728 | 2048.0 | 0.3881 | 0.0027 | 0.0025 | 5.5296 | 2048.0 |
| PROTECTED | 0.3693 | 0.0028 | 0.0025 | 7.7344 | 2048.0 | 0.3382 | 0.0023 | 0.0020 | 4.7104 | 2048.0 |
| S-MuZero | 0.4748 | 0.0357 | 0.0321 | 7.4599 | 208.4 | 0.4322 | 0.0914 | 0.0823 | 12.9971 | 142.2 |
| S-MuZero-worst | 0.4754 | 0.0396 | 0.0356 | 8.3912 | 211.9 | 0.4470 | 0.1032 | 0.0929 | 15.7277 | 152.4 |
| S-MuZero-random | 0.4764 | 0.0423 | 0.0381 | 8.9634 | 211.9 | 0.4526 | 0.1043 | 0.0939 | 16.2812 | 156.1 |
| RobustZero | 0.4972 | 0.0464 | 0.0418 | 9.9992 | 215.5 | 0.4551 | 0.1102 | 0.0992 | 17.6320 | 160.0 |

| Method | IEEE 34-bus | | | | | IEEE 123-bus | | | | |
|---|---|---|---|---|---|---|---|---|---|---|
| | TT | ST | TeT | STE | NSE | TT | ST | TeT | STE | NSE |
| ATLA-PPO | 0.3823 | 0.0079 | 0.0071 | 16.7524 | 2048.0 | 0.3911 | 0.0131 | 0.0118 | 26.8363 | 2048.0 |
| PROTECTED | 0.3432 | 0.0077 | 0.0067 | 15.7696 | 2048.0 | 0.3683 | 0.0127 | 0.0115 | 26.0096 | 2048.0 |
| S-MuZero | 0.5746 | 0.2031 | 0.1828 | 15.5778 | 76.7 | 0.5848 | 0.7226 | 0.6503 | 54.7008 | 75.7 |
| S-MuZero-worst | 0.5773 | 0.2112 | 0.1901 | 16.2835 | 77.1 | 0.5869 | 0.7466 | 0.6719 | 57.1896 | 76.6 |
| S-MuZero-random | 0.5792 | 0.2182 | 0.1964 | 16.8232 | 77.1 | 0.5893 | 0.7492 | 0.6743 | 57.3887 | 76.6 |
| RobustZero | 0.5854 | 0.2236 | 0.2012 | 17.3514 | 77.6 | 0.5951 | 0.7582 | 0.6824 | 58.8363 | 77.6 |

| Method | IEEE 8500-node | | | | | Highway | | | | |
|---|---|---|---|---|---|---|---|---|---|---|
| | TT | ST | TeT | STE | NSE | TT | ST | TeT | STE | NSE |
| ATLA-PPO | 0.5012 | 0.0210 | 0.0191 | 43.0080 | 2048.0 | 0.4208 | 0.0092 | 0.0085 | 18.8416 | 2048.0 |
| PROTECTED | 0.4845 | 0.0201 | 0.0181 | 41.1648 | 2048.0 | 0.4116 | 0.0089 | 0.0080 | 18.2272 | 2048.0 |
| S-MuZero | 0.5983 | 1.0021 | 0.9019 | 76.6606 | 76.5 | 0.5725 | 0.3057 | 0.2751 | 23.6612 | 77.4 |
| S-MuZero-worst | 0.6027 | 1.0335 | 0.9302 | 79.6334 | 77.1 | 0.5736 | 0.3120 | 0.2808 | 24.2112 | 77.6 |
| S-MuZero-random | 0.6011 | 1.0342 | 0.9308 | 78.2158 | 75.6 | 0.5758 | 0.3135 | 0.2822 | 24.4217 | 77.9 |
| RobustZero | 0.6120 | 1.0718 | 0.9646 | 81.7577 | 76.3 | 0.5814 | 0.3218 | 0.2896 | 25.1326 | 78.1 |

| Method | Intersection | | | | | Racetrack | | | | |
|---|---|---|---|---|---|---|---|---|---|---|
| | TT | ST | TeT | STE | NSE | TT | ST | TeT | STE | NSE |
| ATLA-PPO | 0.4604 | 0.0115 | 0.0108 | 23.5520 | 2048.0 | 0.4104 | 0.0080 | 0.0074 | 16.3840 | 2048.0 |
| PROTECTED | 0.4415 | 0.0112 | 0.0101 | 22.9376 | 2048.0 | 0.4015 | 0.0079 | 0.0071 | 16.1792 | 2048.0 |
| S-MuZero | 0.5814 | 0.3848 | 0.3463 | 30.0145 | 78.0 | 0.5623 | 0.2856 | 0.2570 | 21.9916 | 77.0 |
| S-MuZero-worst | 0.5933 | 0.3910 | 0.3519 | 30.5918 | 78.2 | 0.5644 | 0.2903 | 0.2613 | 22.6455 | 78.0 |
| S-MuZero-random | 0.5925 | 0.3922 | 0.3530 | 30.4982 | 77.8 | 0.5632 | 0.2915 | 0.2624 | 22.5532 | 77.4 |
| RobustZero | 0.6038 | 0.4015 | 0.3614 | 31.1163 | 77.5 | 0.5809 | 0.3038 | 0.2734 | 24.0248 | 79.1 |

| Method | Hopper | | | | | Walker2d | | | | |
|---|---|---|---|---|---|---|---|---|---|---|
| | TT | ST | TeT | STE | NSE | TT | ST | TeT | STE | NSE |
| ATLA-PPO | 0.3010 | 0.0014 | 0.0011 | 2.8769 | 2048.0 | 0.5098 | 0.0022 | 0.0019 | 4.4901 | 2048.0 |
| PROTECTED | 0.3225 | 0.0013 | 0.0009 | 2.7684 | 2048.0 | 0.4795 | 0.0018 | 0.0016 | 3.6420 | 2048.0 |
| S-MuZero | 0.3949 | 0.0738 | 0.0685 | 11.1581 | 151.3 | 0.5471 | 0.1126 | 0.1034 | 17.1104 | 152.0 |
| S-MuZero-worst | 0.4066 | 0.0767 | 0.0742 | 11.6543 | 152.0 | 0.5560 | 0.1195 | 0.1086 | 18.2534 | 152.8 |
| S-MuZero-random | 0.4022 | 0.0751 | 0.0733 | 11.4850 | 151.8 | 0.5584 | 0.1213 | 0.1105 | 18.4677 | 152.3 |
| RobustZero | 0.4168 | 0.0816 | 0.0798 | 12.3549 | 151.5 | 0.5678 | 0.1325 | 0.1210 | 20.2796 | 153.1 |

| Method | HalfCheetah | | | | | Ant | | | | |
|---|---|---|---|---|---|---|---|---|---|---|
| | TT | ST | TeT | STE | NSE | TT | ST | TeT | STE | NSE |
| ATLA-PPO | 0.5106 | 0.0023 | 0.0021 | 4.5825 | 2048.0 | 0.5489 | 0.0029 | 0.0027 | 5.9043 | 2048.0 |
| PROTECTED | 0.4821 | 0.0019 | 0.0017 | 3.8054 | 2048.0 | 0.5374 | 0.0028 | 0.0026 | 5.6511 | 2048.0 |
| S-MuZero | 0.5517 | 0.1146 | 0.1082 | 17.5410 | 153.1 | 0.5712 | 0.1741 | 0.1609 | 26.7350 | 153.6 |
| S-MuZero-worst | 0.5608 | 0.1260 | 0.1168 | 19.3452 | 153.5 | 0.5810 | 0.1764 | 0.1668 | 27.1345 | 153.8 |
| S-MuZero-random | 0.5647 | 0.1287 | 0.1175 | 19.7628 | 153.6 | 0.5847 | 0.1798 | 0.1679 | 27.6278 | 153.7 |
| RobustZero | 0.5782 | 0.1376 | 0.1232 | 21.1976 | 154.0 | 0.5917 | 0.1850 | 0.1706 | 28.4967 | 154.1 |

while the worst-case reward increases.

- **Impact of $\lambda_2$.** Fig. 4 reports rewards obtained by RobustZero when varying $\lambda_2$. We observe that all rewards first increase and then decrease as $\lambda_2$ decreases. To understand why, note that $\lambda_2$ is used to control the ratio of the magnitudes between $w2_{t'}\theta_{t'-1}$ and $\nabla\widetilde{\mathcal{L}}_{\theta_{t'-1}} + w1_{t'}\nabla\hat{\mathcal{L}}_{\theta_{t'-1}}$ (see Eq. 9). On the one hand, a too large $\lambda_2$ means that $w2_{t'}\theta_{t'-1}$ maintains a high value during training. This will greatly reduce the impact of the worst-case and random-case loss terms, resulting in reduced natural, worst-case, and random-case rewards. On the other hand, if $\lambda_2$ is too small, $w2_{t'}$ remains too small. This leads to reduced adjustment functionality of $w2_{t'}$, which also results in reduced natural, worst-case, and random-case rewards. Based on these analyses, the settings of both large and small $\lambda_2$ are not beneficial for achieving high rewards.

*Table 8.* The relationship between environment samples and rewards on CartPole, Pendulum, IEEE 34-bus, IEEE 123-bus, IEEE 8500-node, Highway, Intersection, and Racetrack.

| Method | Pendulum | | | | Cartpole | | | |
|---|---|---|---|---|---|---|---|---|
| | Samples | Natural Reward | Worst-case Reward | Random-case Reward | Samples | Natural Reward | Worst-case Reward | Random-case Reward |
| RobustZero | 73,776 | -110.48 | -116.61 | -110.49 | 69,632 | 474.60 | 438.09 | 471.84 |
| | 147,552 | -89.98 | -111.00 | -89.03 | 40,261 | 496.09 | 456.32 | 487.27 |
| | 221,328 | -83.99 | -88.15 | -84.98 | 210,891 | 500.00 | 459.33 | 489.56 |
| ATLA-PPO | 73,728 | -112.79 | -167.42 | -100.90 | 69,632 | 384.66 | 325.31 | 390.76 |
| | 147,456 | -92.02 | -95.18 | -95.64 | 40,960 | 474.60 | 356.09 | 371.84 |
| | 221,184 | -92.18 | -95.34 | -94.62 | 212,992 | 477.94 | 369.32 | 375.46 |
| PROTECTED | 73,728 | -119.59 | -117.80 | -105.36 | 69,632 | 457.19 | 376.10 | 380.54 |
| | 147,456 | -94.33 | -104.32 | -100.79 | 40,960 | 462.52 | 428.36 | 418.76 |
| | 221,184 | -93.86 | -99.32 | -96.52 | 212,992 | 465.86 | 421.69 | 422.09 |

| Method | IEEE 34-bus | | | | IEEE 123-bus | | | |
|---|---|---|---|---|---|---|---|---|
| | Samples | Natural Reward | Worst-case Reward | Random-case Reward | Samples | Natural Reward | Worst-case Reward | Random-case Reward |
| RobustZero | 177,383 | -21.29 | -29.44 | -21.38 | 198,042 | -25.29 | -20.89 | -24.74 |
| | 354,765 | -13.31 | -18.83 | -17.79 | 396,084 | -13.50 | -16.39 | -14.72 |
| | 532,148 | -9.78 | -12.88 | -11.76 | 594,128 | -10.99 | -13.85 | -12.95 |
| ATLA-PPO | 178,176 | -25.47 | -26.08 | -23.92 | 198,656 | -36.30 | -22.56 | -25.44 |
| | 354,304 | -21.29 | -24.44 | -28.38 | 395,264 | -25.29 | -20.89 | -22.74 |
| | 532,480 | -17.16 | -18.31 | -15.56 | 594,128 | -21.32 | -18.16 | -18.44 |
| PROTECTED | 178,176 | -21.19 | -20.51 | -30.83 | 198,656 | -34.85 | -26.81 | -28.62 |
| | 354,304 | -13.80 | -16.29 | -24.99 | 395,264 | -29.93 | -22.21 | -27.11 |
| | 532,480 | -12.62 | -15.38 | -18.99 | 594,128 | -19.93 | -15.21 | -17.11 |

| Method | IEEE 8500-node | | | | Racetrack | | | |
|---|---|---|---|---|---|---|---|---|
| | Samples | Natural Reward | Worst-case Reward | Random-case Reward | Samples | Natural Reward | Worst-case Reward | Random-case Reward |
| RobustZero | 271,413 | -1350 | -1567 | -1421 | 138,064 | 498.65 | 445.35 | 470.21 |
| | 542,827 | -1255 | -1475 | -1315 | 276,129 | 507.36 | 463.20 | 479.19 |
| | 814,240 | -1175 | -1386 | -1225 | 414,193 | 519.30 | 473.22 | 485.30 |
| ATLA-PPO | 272,384 | -1454 | -1769 | -1506 | 137,216 | 364.74 | 326.57 | 348.37 |
| | 542,720 | -1350 | -1667 | -1421 | 276,480 | 385.72 | 349.71 | 363.36 |
| | 815,104 | -1358 | -1574 | -1417 | 413,696 | 381.32 | 361.84 | 371.84 |
| PROTECTED | 272,384 | -1763 | -1888 | -1734 | 137,216 | 374.17 | 338.74 | 349.86 |
| | 542,720 | -1457 | -1671 | -1427 | 276,480 | 411.04 | 354.74 | 359.71 |
| | 815,104 | -1358 | -1574 | -1417 | 413,696 | 417.32 | 361.28 | 371.28 |

| Method | Highway | | | | Intersection | | | |
|---|---|---|---|---|---|---|---|---|
| | Samples | Natural Reward | Worst-case Reward | Random-case Reward | Samples | Natural Reward | Worst-case Reward | Random-case Reward |
| RobustZero | 71,594 | 15.42 | 15.38 | 13.77 | 127,725 | 2.47 | 2.35 | 2.41 |
| | 143,188 | 21.98 | 17.15 | 17.02 | 255,449 | 2.86 | 2.44 | 2.66 |
| | 214,782 | 23.10 | 20.33 | 21.95 | 383,174 | 3.21 | 2.75 | 3.01 |
| ATLA-PPO | 71,680 | 13.31 | 12.98 | 6.86 | 126,976 | 1.62 | 0.96 | 1.02 |
| | 143,360 | 15.66 | 14.54 | 10.86 | 256,000 | 1.97 | 1.28 | 1.61 |
| | 215,040 | 16.23 | 15.98 | 13.33 | 382,976 | 2.03 | 1.33 | 1.63 |
| PROTECTED | 71,680 | 11.19 | 14.37 | 15.07 | 126,976 | 1.11 | 0.86 | 0.93 |
| | 143,360 | 13.02 | 15.69 | 16.75 | 256,000 | 1.65 | 1.30 | 1.36 |
| | 215,040 | 13.63 | 17.38 | 18.13 | 382,976 | 2.18 | 1.38 | 1.62 |

## C.8. Impact of The Attack Radius $\epsilon$

We evaluate the performance of RobustZero and all baselines under varying values of attack radius $\epsilon$. RobustZero does not know the attack budget $\epsilon$ during training. Figs. 5–9 illustrate the changes in natural rewards, B1-worst-case rewards, B2-worst-case rewards, U-random-case rewards, and G-random-case rewards as the increase of $\epsilon$ across all environments. The key observations are as follows: 1) The natural rewards of S-MuZero remain unaffected by changes in $\epsilon$, as it is trained without state perturbations. Nevertheless, its B1-worst-case rewards, B2-worst-case rewards, U-random-case rewards, and G-random-case rewards decrease dramatically. For all other methods, natural rewards decrease as $\epsilon$ increases. Note that RobustZero achieves higher natural rewards than ATLA-PPO, PROTECTED, S-MuZero-worst, and S-MuZero-random across all $\epsilon$ settings and environments. The exception is on CartPole, where RobustZero, ATLA-PPO, PROTECTED,

*Table 9.* The relationship between environment samples and rewards on Hopper, Walker2d, HalfCheetah, and Ant.

| Method | **Hopper** | | | | | **Walker2d** | | | |
|---|---|---|---|---|---|---|---|---|---|
| | Samples | Natural Reward | Worst-case Reward | Random-case Reward | Samples | Natural Reward | Worst-case Reward | Random-case Reward | |
| RobustZero | 333,521 | 2937 | 2066 | 2848 | 661,089 | 5150 | 4957 | 5112 | |
| | 667,042 | 3418 | 2424 | 3293 | 1,322,179 | 5902 | 5717 | 5912 | |
| | 1,000,564 | 3610 | 2554 | 3517 | 1,983,269 | 6308 | 6075 | 6287 | |
| ATLA-PPO | 331,776 | 2164 | 1136 | 2079 | 661,504 | 2556 | 2393 | 2507 | |
| | 665,600 | 2802 | 1497 | 2696 | 1,323,008 | 3344 | 3180 | 3294 | |
| | 999,424 | 2945 | 1544 | 2824 | 1,982,464 | 3483 | 3281 | 3465 | |
| PROTECTED | 331,776 | 2389 | 1658 | 2303 | 661,504 | 4158 | 3972 | 4070 | |
| | 665,600 | 3175 | 2198 | 3050 | 1,323,008 | 5562 | 5210 | 5315 | |
| | 999,424 | 3280 | 2267 | 3182 | 1,982,464 | 5738 | 5413 | 5540 | |
| Method | **HalfCheetah** | | | | | **Ant** | | | |
| | Samples | Natural Reward | Worst-case Reward | Random-case Reward | Samples | Natural Reward | Worst-case Reward | Random-case Reward | |
| RobustZero | 710,936 | 5711 | 4357 | 5099 | 800,568 | 4670 | 3789 | 4591 | |
| | 1,421,872 | 6591 | 5024 | 5883 | 1,601,137 | 5345 | 4390 | 5325 | |
| | 2,132,808 | 7060 | 5345 | 6320 | 2,401,706 | 5721 | 4636 | 5615 | |
| ATLA-PPO | 710,656 | 4030 | 3346 | 3987 | 800,768 | 3486 | 2438 | 3473 | |
| | 1,421,312 | 5338 | 4407 | 5310 | 1,601,536 | 4628 | 3246 | 4620 | |
| | 2,131,968 | 5561 | 4616 | 5500 | 2,402,304 | 4802 | 3365 | 4786 | |
| PROTECTED | 710,656 | 4675 | 3375 | 4090 | 800,768 | 3799 | 3010 | 3696 | |
| | 1,421,312 | 6090 | 4428 | 5379 | 1,601,536 | 5033 | 3932 | 4857 | |
| | 2,131,968 | 6377 | 4641 | 5652 | 2,402,304 | 5200 | 4107 | 5073 | |

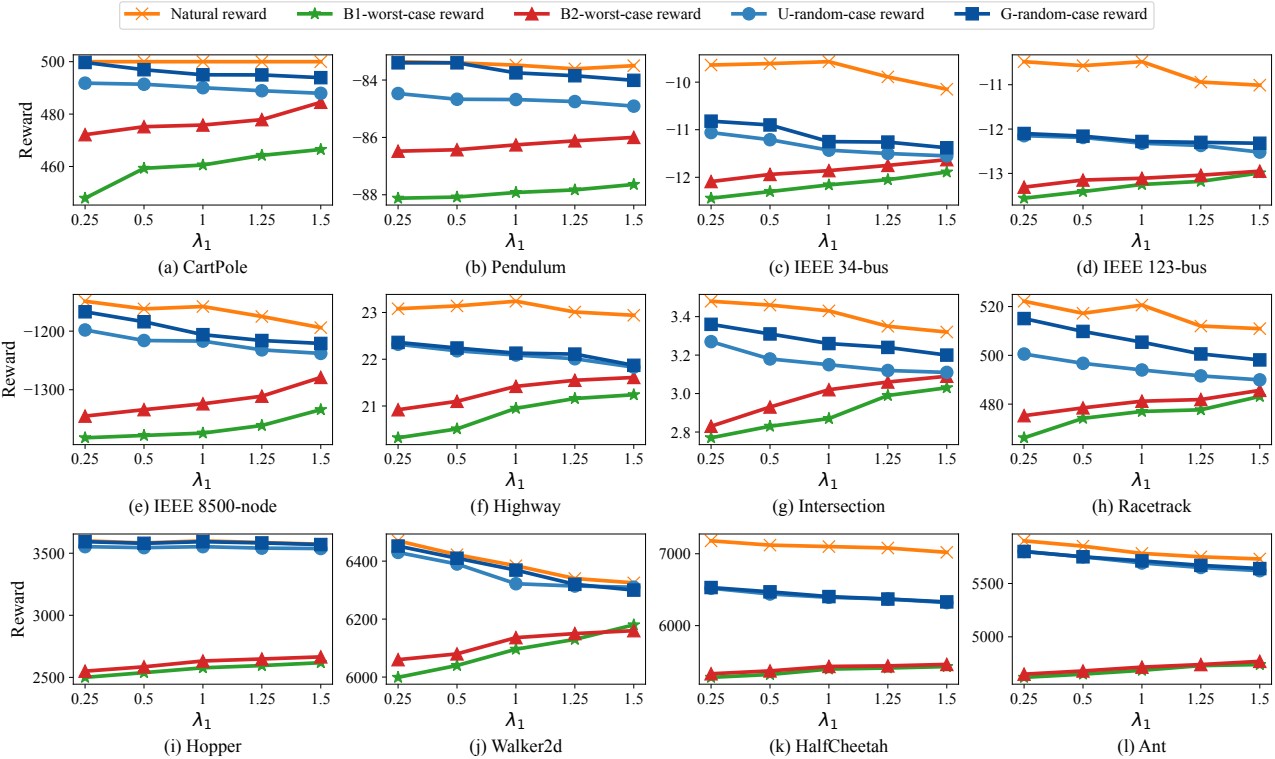

*Figure 3.* Performance of RobustZero with different $\lambda_1$ across all environments.

and S-MuZero consistently attain optimal natural rewards, regardless of $\epsilon$; and 2) B1-worst-case rewards, B2-worst-case rewards, U-random-case rewards, and G-random-case rewards for all methods decline as $\epsilon$ increases. This indicates that larger attack radius weaken the defense capabilities of all methods to state perturbations. This is because as $\epsilon$ increases, attacks become stronger. This further increases the difficulty to defend against such attacks. Despite this, RobustZero consistently outperforms all baselines, which achieves the highest worst-case and random-case rewards across all $\epsilon$ values and environments. These results imply that RobustZero is consistently the best option across all $\epsilon$ settings.

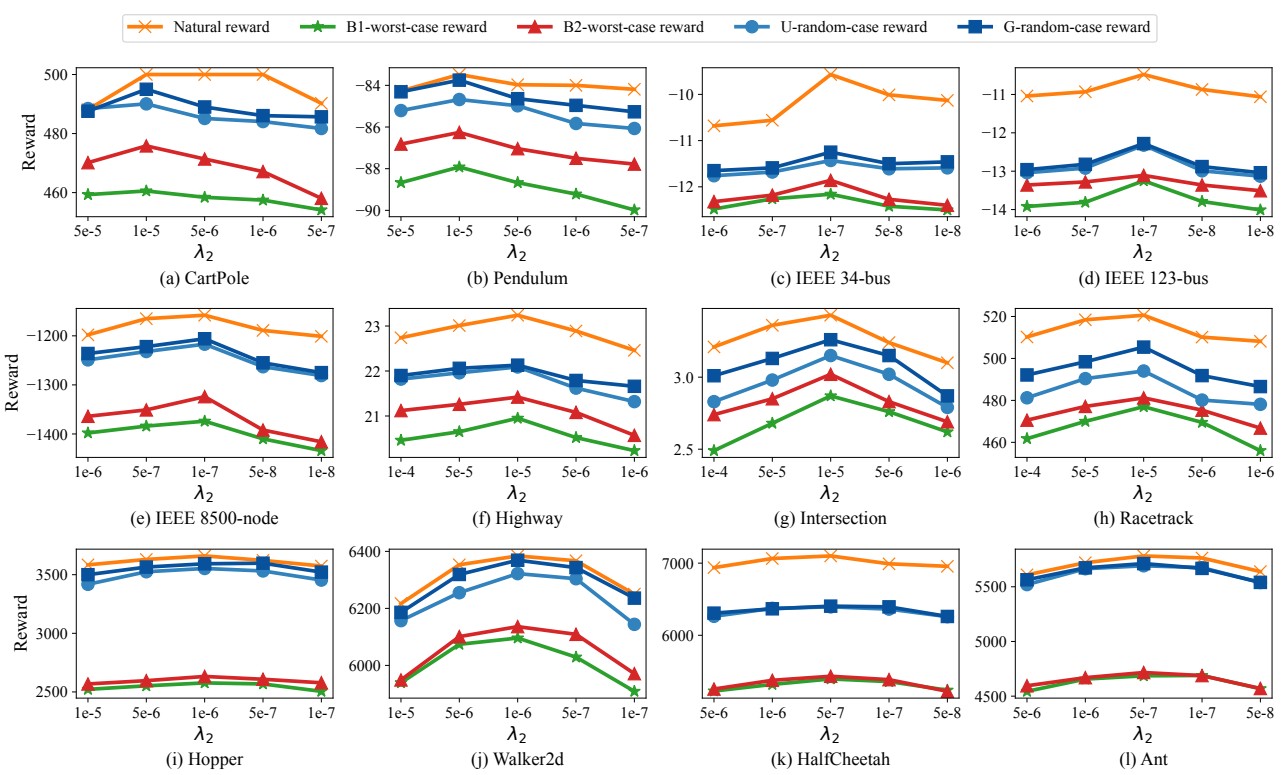

*Figure 4.* Performance of RobustZero with different $\lambda_2$ across all environments.

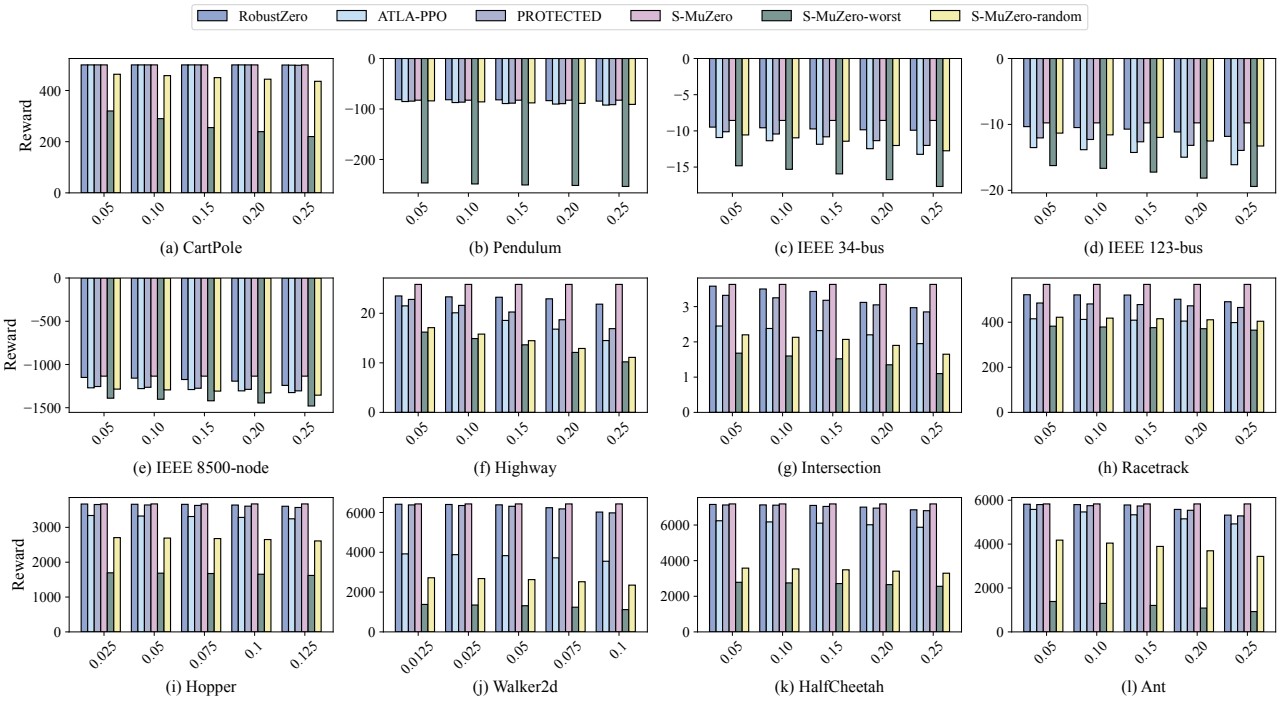

*Figure 5.* Natural rewards of RobustZero and baselines with different values of $\epsilon$ across all environments.

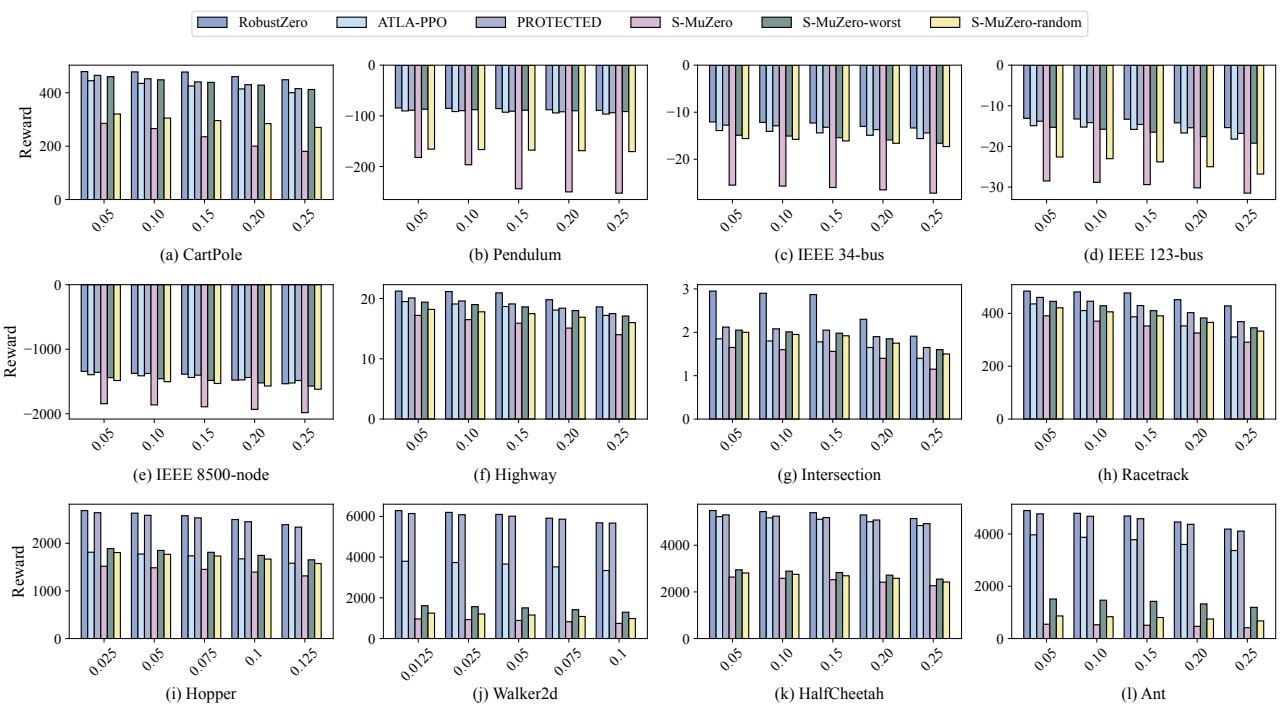

*Figure 6.* B1-worst-case rewards of RobustZero and baselines with different values of $\epsilon$ across all environments.

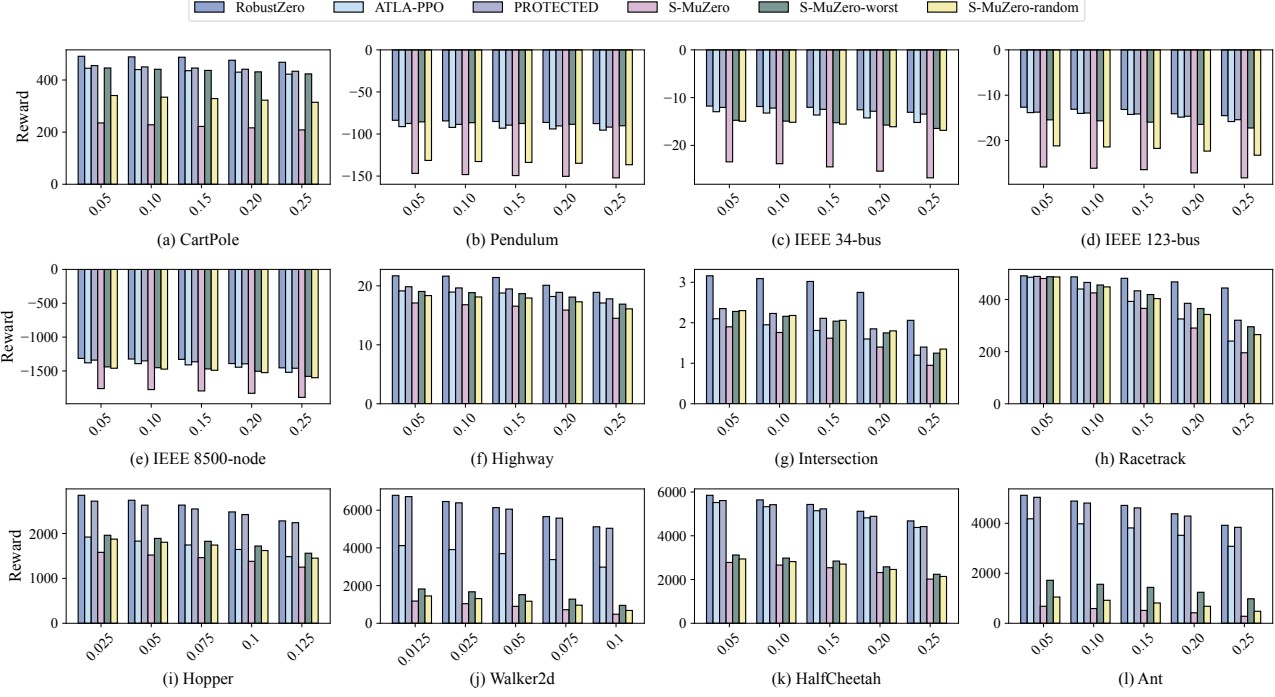

*Figure 7.* B2-worst-case rewards of RobustZero and baselines with different values of $\epsilon$ across all environments.

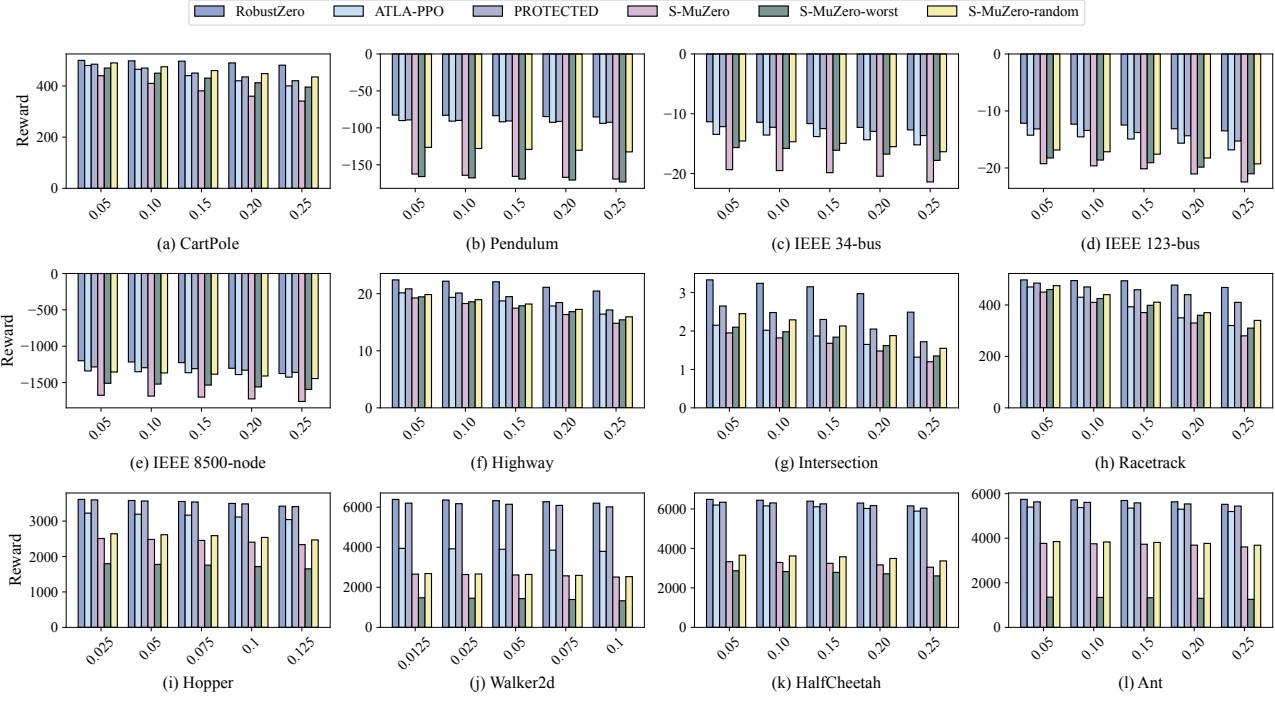

*Figure 8.* U-random-case rewards of RobustZero and baselines with different values of $\epsilon$ across all environments.

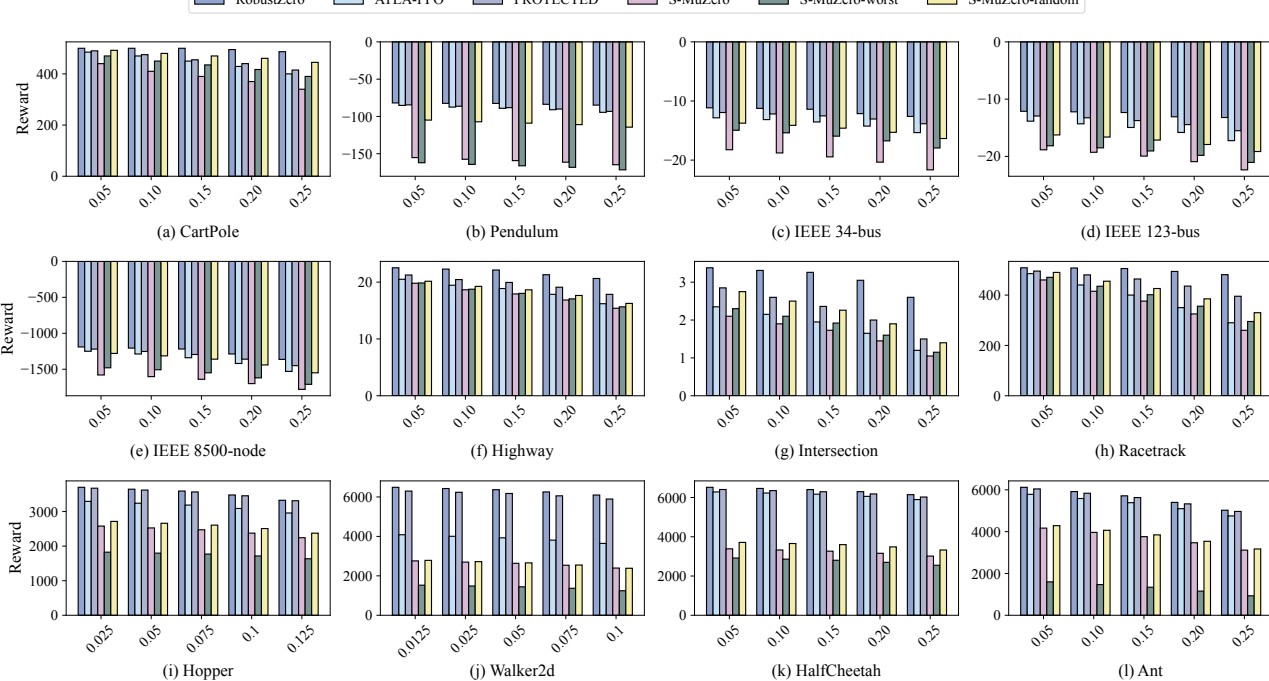

*Figure 9.* G-random-case rewards of RobustZero and baselines with different values of $\epsilon$ across all environments.

