# OpenReview forum: "RobustZero: Enhancing MuZero Reinforcement Learning Robustness to State Perturbations"
_ICML.cc/2025/Conference — ICML 2025 poster_

### Official Review · Reviewer_ZVib · 2025-03-07

**Overall Recommendation:** 4

**Summary:**

The paper adapts the MuZero algorithm (RobustZero) with a state-robustness loss and two adaptive hyper-parameter adjustment methods. The method is designed to deal with both worst-case and random perturbations of the state, if those are available during training. The authors evaluate their approach against the S-MuZero baseline and the ATLA-PPO and PROTECTED algorithms on 4 reasonable benchmark tasks. While RobustZero does not perform in all tasks significantly better than the baselines, it shows significant improvement in the IEEE tasks, in particular under "random perturbation", and generally in the perturbed RaceTrack environment. The authors also perform a large number of ablations, which show that their method with adjustment between worst-case and random-case perturbation outperforms robustifying only one of these cases, and how large the influence of the proposed changes to S-MuZero are.

**Claims And Evidence:**

- "The presented method is the first that achieves robustness against state perturbation with MuZero." I am not an expert in robust RL, but I have not seen any relevant MuZero papers claiming to do that, so I am inclined to believe this claim.
- "RobustZero performs well in both worst-case and random-case state perturbation." While no strictly true for all environments, RobustZero seems to outperform the baselines in both perturbed cases on Racetrack and in the randomly-perturbed case in the two IEEE environments. Somewhat surprisingly, even baselines that specifically optimize for that type of perturbation are outperformed here, which is pretty impressive.

In summary, I believe the authors back their claims up sufficiently.

**Essential References Not Discussed:**

None.

**Experimental Designs Or Analyses:**

There is are a decent amount of ablations, demonstrating the usefulness of the proposed changes, which is welcome. Making **all** significantly better entries in bold will also enhance this Table.

**Methods And Evaluation Criteria:**

The method seems to be mostly adapted from Liu et al. (2024), and its implementation is not terribly surprising. However, it is also not trivial and passes my personal threshold for sufficient novelty in an ICML paper. Which is of course subjective.

I did not fully understand why RobustZero requires projector and predictor networks. I have seen similar arrangements when transition models are learned, and the encoding of $s_{t+1}$ can therefore be predicted from a projection of the encoding of $s_t$, but here this approach seems meaningless. It is also not clear why the projection does not simply learn a constant output (which would be guaranteed to be the same under perturbation). The authors claim that "the key functionality of the predictor is to transform projected features to stabilize optimization and reduce collapsing solutions", but I do not see why that is. Do the authors have a better explanation for this?

Evaluation seems to be sufficient, with two suggestions for the authors:
- Please do not make only the *best* entry in a column bold, this can easily deceive a casual reader. Make **all** entries bold that are not significantly worse (overlapping standard deviations) than then best entry. In your case the Pendulum task is practically useless, and some columns in the IEEE benchmarks should have multiple bold entries.
- I am missing learning curves (performance in all three reward cases over environment steps). These are technically not necessary, but as a reinforcement learner I always want to see them to make sure the results are actually stable and the final results are not cherry picked. Please add some to the appendix.

**Other Comments Or Suggestions:**

- l.36R: Your method does not address "Even a small reality gap can compound errors in the learned models", as you only consider robust state encoding, which does not compound (only future predictions do this). Please clarify here which error you are addressing in the paper.
- l.135L: the MDP is missing an initial-state distribution. You should also define $\mathcal P$ more precisely: is it stochastic or deterministic? Also, please mention that your state space must be continuous (to allow $\epsilon$-balls).
- l.142L+157L+115R: Please denote $o_t$ *either* a state *or* as an observation, not both. I recommend "state" to differentiate it from the observations in POMDP.
- l.117R: when you define the goal of MuZero, please clarify which action sequence has been used to compute, e.g., $p_t^k$.
- l.144Lf: the goal is to maximize the *expected* return.
- l.161L+163L: the reward $r_t^k$ is usually the output of $\mathcal F$, not $\mathcal G$ in MuZero.
- l.141R: "environment model" -> "MDP"
- l.154R: The sentence "$\tilde t$ and $\hat t$ denote time under worst-case and random-case state perturbations" makes no sense. Time is not perturbed. You mean to say that $\tilde t$ denotes data at time $t$ from a trajectory that is drawn under worst-case perturbation.
- Please add the mathematical denotations of the networks in Figure 1.
- Define $\mathcal D_{WC}$ and $\mathcal D_{RC}$ somewhere, e.g., in Figure 1.
- In any case, it must be $\mathcal B \sim \mathcal D_{WC}$, not the other way around, and $\mathcal B$ is the batch and not the batch size (l.249L).
- eq.10 misses a $||_2$

**Other Strengths And Weaknesses:**

The paper is mostly well written, but sometimes, in particular in the formal parts, it is a bit hard to understand.

**Questions For Authors:**

- Can the authors explain why "the predictor is to transform projected features" leads to "stabiliz[ation of] optimization and reduc[tion] collapsing solutions"?
- Why do the authors not also use a third MuZero loss that optimizes for the unperturbed states? RobustZero already uses them and a third learning signal might improve performance further.

**Relation To Broader Scientific Literature:**

I am not an expert in robust RL, but the MuZero literature is sufficiently covered.

**Theoretical Claims:**

None.

---

> ### Author Rebuttal · Authors · 2025-03-31
>
> We appreciate your positive and valuable comments.
>
> $\textbf{Response to Methods and Evaluation Criteria:}$ 1) Regarding the projector and predictor networks, please refer to the response to Q1; 2) Regarding the bold entries, we have revised all tables (see https://anonymous.4open.science/r/RobustZero-SupportMaterials-8512/Tables/Tables%201-5.png); 3) We have added the learning curves (see https://anonymous.4open.science/r/RobustZero-SupportMaterials-8512/Figures/Figure%20S-1.png).
>
> $\textbf{Response to Experimental Designs or Analyses:}$ We have revised all tables according to your suggestion.
>
> $\textbf{Response to Other Comments or Suggestions:}$ We would like to address your comments as follows:
>
> 1)  MuZero-class methods learn abstract environment models, i.e., learned models, and employ MCTS for planning. At each time step, a real state is mapped to an abstract initial state by using the representation network. Then, the dynamics network and prediction network work in conjunction with MCTS to predict future evolution of these abstract states. This framework inherently involves a prediction process.
> The compound errors mean the accumulation of inaccuracies in these predicted abstract states. To clarify this point, we will revise the statement to ‘Even a small “reality gap” can compound errors in the predicted abstract states within MuZero-class methods, causing reduced rewards and potentially harmful decision-making’.
>
> 2) We will supplement the following missing information: i) $\rho_0$ is the initial state distribution; ii) $\mathcal{P}$ is a deterministic transition function; and 3) the state space is continuous.
>
> 3) We will use “state” uniformly.
>
> 4) We will add the following statement. The sequence of real actions from the sampled trajectory is used to compute $r_t^k$.
>
> 5) We will revise the statement to “the goal is to maximize the expected return.”
>
> 6) We have carefully double-checked the MuZero algorithm, and make sure that the reward  $r_t^k$ is the output of the dynamics network $\mathcal{G}$.
>
> 7) We will replace “environment model” with “MDP”.
>
> 8) We will revise the statement to “$\widetilde{t}$ and $\hat{t}$ denote data at time t from a trajectory generated under worst-case and random perturbations, respectively.”
>
> 9) We have revised Fig. 1 by adding the mathematical denotations of the networks (see https://anonymous.4open.science/r/RobustZero-SupportMaterials-8512/Figures/Figure%201.png).
>
> 10) We have defined $\mathcal{D}\_{\textit{WC}}$  and $\mathcal{D}\_{\textit{RC}}$ in Fig. 1 (see the link above).
>
> 11) We will revise the expressions: i) From $ \mathcal{D}\_{\textit{WC}}\sim\mathcal{B}$ to $ \mathcal{B}\sim\mathcal{D}\_{\textit{WC}}$; ii) From $ \mathcal{D}\_{\textit{RC}}\sim\mathcal{B}$ to $ \mathcal{B}\sim\mathcal{D}\_{\textit{RC}}$; and iii) From “B is the batch size” to  “B is the batch”.
>
> 12)  We will add the missing symbol for Eq. 10.
>
> $\textbf{Response to Q1:}$  Model collapse refers to a degenerate solution in which the encoder produces identical or nearly identical outputs for all inputs, resulting in trivial and non-informative representations. If we omit the predictor and directly enforce similarity between the outputs of two branches (e.g., using mean squared error or cosine similarity), the model may exploit a shortcut: “I might as well just output a constant vector, since all inputs yield the same output, the loss will be minimized.” This leads to model collapse. When collapse occurs, all states—regardless of whether they are perturbed or not—are mapped to the same or very similar initial hidden state, effectively losing the ability to distinguish between different inputs.  To avoid model collapse, one effective strategy is to introduce an asymmetric network architecture, where a predictor is added to one branch, while the other branch remains without it and has its gradient flow stopped. This asymmetry prevents both branches from trivially converging to the same constant output. The branch with the predictor learns to align its output with the target branch, which acts as a stable reference point. Because the target branch does not receive gradients, it cannot adjust itself to match the predictor’s potentially trivial solution, thus breaking the symmetry that often leads to collapse. This design encourages the network to learn meaningful representations, resulting in stabilization of optimization and a reduction in collapsing solutions.
>
> $\textbf{Response to Q2:}$ The reasons are as follows. First, as shown in Eqs. 3-4, both the worst-case and random-case loss terms already incorporate information from perturbed and unperturbed states due to the use of a contrastive loss function. This contrastive loss encourages the learned policy to remain consistent before and after state perturbations. Therefore, an additional MuZero loss term specifically targeting unperturbed states is redundant. Second, an additional loss term will increase the computational cost.

---

### Official Review · Reviewer_v7JN · 2025-03-10

**Overall Recommendation:** 3

**Summary:**

This work proposed a robust version of Muzero framework called RobustZero to gain robustness when facing state perturbations. Muzero features a self-supervised representation network to generate a consistent initial hidden state and a unique loss function to gain robustness. In the experiment setting, Muzero shows superior performance under both worst and random state perturbations.

## update after rebuttal

The rebuttal has adequately addressed most of my concerns, and I am now leaning toward accepting this paper.

**Claims And Evidence:**

I believe the claims are well supported.

**Essential References Not Discussed:**

I believe there are recent works in this field using diffusion models to gain robustness under state perturbation attacks that should be cited.

[1] DMBP: Diffusion model based predictor for robust offline reinforcement learning against state observation perturbations. ICLR 2024
[2] Belief-Enriched Pessimistic Q-Learning against Adversarial State Perturbations. ICLR 2024

**Experimental Designs Or Analyses:**

The evaluation of the proposed method is similar to previous related work by providing rewards under no attack and rewards under worst and random perturbations. However, I do have some concerns about the experiments. 1. The environments do not include Mojuco, which is a commonly used environment in previous related work. 2. The worst attack is achieved by using ATLA-PPO, why not use PA-AD[1] attack that is currently considered the strongest attack? 3. It would be better to include some classical baselines that are compared in most related work, such as SA-PPO[2] in the experiments.

[1] Who Is the Strongest Enemy? Towards Optimal and Efficient Evasion Attacks in Deep RL. ICLR 2022
[2] Robust Deep Reinforcement Learning against Adversarial Perturbations on State Observations NeurIps 2020

**Methods And Evaluation Criteria:**

Yes, the proposed methods gain the best robustness under both worst-case and random perturbation under various environments.

**Other Comments Or Suggestions:**

N/A

**Other Strengths And Weaknesses:**

Strengths:
1. Comprehensive experiment results and alation studies reported by the authors.
2. The first work to consider state perturbation in Muzero algorithm.

Weaknesses:
1. There are multiple hyperparameters introduced in the Robustzero, and it is shown in the Appendix that the optimal parameters are different across different environments. Does Robustzero need to search for the best parameters for a new environment?

**Questions For Authors:**

I will summarize all my questions here.
1. The environments do not include Mojuco, which is a commonly used environment in previous related work.
2. The worst attack is achieved by using ATLA-PPO, why not use the PA-AD[1] attack that is currently considered the strongest attack?
3. It would be better to include some classical baselines that are compared in most related work, such as SA-PPO[2] in the experiments.
4. There are multiple hyperparameters introduced in the Robustzero, and it is shown in the Appendix that the optimal parameters are different across different environments. Does Robustzero need to search for the best parameters for a new environment?
5. Does Robustzero know the attack budget $\epsilon$ during training? If it knows, how is the performance with uncertain attack budget $\epsilon$?
6. Diffusion model based methods also achieved strong robustness, as shown in [3]. Could the authors include it as a baseline? I believe the setting of [3] is similar to the environments used in the paper.
7. Could the author provide training time and testing time comparisons?

[1] Who Is the Strongest Enemy? Towards Optimal and Efficient Evasion Attacks in Deep RL. ICLR 2022

[2] Robust Deep Reinforcement Learning against Adversarial Perturbations on State Observations NeurIps 2020

[3] DMBP: Diffusion model based predictor for robust offline reinforcement learning against state observation perturbations. ICLR 2024

**Relation To Broader Scientific Literature:**

This work is related to robust RL, trustworthy AI and AI safety in general.

**Theoretical Claims:**

Yes, I checked the formation of the loss functions proposed by the authors.

---

> ### Author Rebuttal · Authors · 2025-03-31
>
> We appreciate your valuable comments and recognition of our contributions.
>
> $\textbf{Response to Experimental Designs or Analyses:}$  Please refer to the responses to Q1-Q3.
>
> $\textbf{Response to Essential References Not Discussed:}$ We will add the two references as follows.  The recent studies [1-2] introduce the diffusion models to enhance robustness to state perturbations. Specifically, a diffusion model-based predictor is proposed [1] for offline RL to recover the actual states against state perturbations. A belief-enriched pessimistic Q-learning method is proposed [2] by using diffusion model to purify observed states.
>
> $\textbf{Response to Q1:}$ Following your suggestion, we have studied RobustZero and all baselines on five Mujoco environments, including Hopper, Walker2d, HalfCheetah, Ant and Humanoid. The results are provided in Table S-1 (see https://anonymous.4open.science/r/RobustZero-SupportMaterials-8512/Tables/Table%20S-1.png). The results show consistently that RobustZero outperforms all baselines at defending against state perturbations. This provides evidence that RobustZero  performs well in general and challenging Mujoco environments.
>
> $\textbf{Response to Q2:}$ We fully agree with the reviewer that PA-AD (referred to as PA-ATLA-PPO in our paper) is currently considered the strongest attack method. However, as mentioned in Section 2.2, it is a white-box attack, requiring access to the internal parameters of the victim model. In contrast, our work focuses on black-box attack settings, where such access is not available. Therefore, we do not include PA-ATLA-PPO as a baseline in our experiments.
>
> $\textbf{Response to Q3:}$  Since SA-PPO is a white-box attack strategy, we do not use it as a baseline.
>
> $\textbf{Response to Q4 and W1:}$  Our method involves four parameters: $w1$, $w2$, $\lambda_1$, and $\lambda_2$. We clarify their design and selection: 1) One of our contributions is the development of an adaptive mechanism to adjust $w1$  and $w2$, eliminating the need for ad-hoc adjustments. Note that in Appendix C.3, we show the selection of $w2$ across different environments. This is used in ablation studies, where we intentionally fix $w2$ and find best value to isolate its effect and demonstrate the advantage of the adaptive mechanism; 2) $\lambda_1$ adjusts the trade-off between robustness to worst-case perturbations and random perturbations. A larger $\lambda_1$ emphasizes worst-case robustness, while a smaller value favors random-case robustness. When both are considered equally important, setting $\lambda_1=1$ (as done in our experiments) is a reasonable and effective default. Thus, $\lambda_1$ does not require tuning unless specific emphasis is desired; and 3) Among the four parameters, $\lambda_2$ is the only one that requires manual tuning. To keep the selection process simple, we employ a standard grid search. Appendix C.7 analyzes the effect of different $\lambda_2$ values, which is not the selection process.  In summary, we only need to search for the best setting of $\lambda_2$  for a new environment. Therefore, the hyperparameter selection is not complex.
>
> $\textbf{Response to Q5:}$ RobustZero does not know the attack budget $\epsilon$ during training. We will clarify this statement in the final paper.
>
>
> $\textbf{Response to Q6:}$ Literature [3] is the first study to design a defense strategy against state perturbations specifically for offline RL. The strength of offline RL lies in its ability to train agents solely from a fixed dataset, without requiring any interaction with the environment. However, this limits its exploration capability, making it more challenging to learn optimal policies. That said, [3] demonstrates strong robustness to state perturbations when compared to other offline RL methods that lack defense strategies. In contrast, RobustZero and all baseline methods evaluated in our work are online RL methods, where the agent learns by actively interacting with the environment. Generally, online RL methods are capable of achieving higher rewards than offline RL methods due to their adaptive exploration. Given the fundamental differences in training protocols and assumptions between offline and online RL, we do not include [3] as a baseline, as it would not constitute a fair comparison under the same experimental setup.
>
> $\textbf{Response to Q7:}$ In Appendix C.6, we provided comparison analysis for the training time and sampling time. Since the sampling time closely approximates the testing time, we initially omitted testing time results in the submitted paper. Now, we have  added the testing time per step (TeT) and updated Table 6 accordingly (refer to https://anonymous.4open.science/r/RobustZero-SupportMaterials-8512/Tables/Table%206.png). As shown in the update Table 6, the testing time is similar to the sampling time, the previous analysis of sampling time remains applicable. For a comprehensive comparison, please refer to Appendix C.6.

---

### Official Review · Reviewer_BDsu · 2025-03-14

**Overall Recommendation:** 3

**Summary:**

The authors propose RobustZero, the first MuZero-class method designed to ensure robustness against state perturbations, including both worst-case and random-case scenarios. The proposed method introduces a training framework that includes a self-supervised representation network, which facilitates the generation of consistent policies both before and after state perturbations. The framework also incorporates a unique loss function that enhances the robustness of the training process. Furthermore, the authors present an adaptive adjustment mechanism that allows for model updates, ensuring high robustness to perturbations. Extensive experiments conducted across eight environments provide strong evidence that RobustZero outperforms existing state-of-the-art methods in defending against state perturbations.

**Claims And Evidence:**

The motivation of this work starts from the limitations of both model-free and model-based methods. Therefore, MuZero was proposed to integrate the strengths from both methods. However, it is not guaranteed that RobustZero will inherit same efficiency benefits compared with robust model-free method. In short, learning curve or other metrics should be included to show that the proposed method is more sample efficient.

**Essential References Not Discussed:**

I will suggest the authors to include the following papers, which are also an important branch of robust RL, especially that it is still not clear what the definition of worst case and random case in this paper's setting. It should be clearly discuss and identify the difference with [2]'s worst/random settings.

[1] Lerrel Pinto et al. "Robust Adversarial Reinforcement Learning", ICML, 2017

[2] Juncheng Dung et al. "Variational Adversarial Training Towards Policies with Improved Robustness", AISTATS, 2025

**Experimental Designs Or Analyses:**

How do you train S-MuZero-worst under worst case? Do you assume that you know the worst case scenario for different policies and is the worst case here fixed during training? This also connects to my another question that how can worst-case reward higher than random-case reward in Racetrack for S-Muzero-worst in table 1? I thought worst case means that the performance should always be the worst.

**Methods And Evaluation Criteria:**

The benchmark is not sufficient. Authors only pick cartpole and pendulum from MuJoco while there are more complicated environments in Mujoco. Only evaluating on more complicated environments can convince that pure model-based methods cannot have the access to prior knowledge of the environment’s dynamics

**Other Comments Or Suggestions:**

* In line 363, I thought it is in each column instead of row.

**Other Strengths And Weaknesses:**

Strengths:
* This paper makes a valuable contribution to the field with promising results.
* Comprehensive analysis and experiments

**Questions For Authors:**

* Can S-MuZero solve the tasks with continuous action space? If yes, the explicit statement should be mentioned in 3.3 section.

**Relation To Broader Scientific Literature:**

Introduce the robustness concept to a blend of model-free and model-based method

**Theoretical Claims:**

No theory

---

> ### Author Rebuttal · Authors · 2025-03-31
>
> We appreciate your positive and valuable comments.
>
> $\textbf{Response to Claims and Evidence:}$ Following your comments, we have analyzed the relationship between the number of environment samples and the natural, worst-case, and random-case rewards for RobustZero and the two robust model-free baselines: ATLA-PPO and PROTECTED. The results are presented in Table S-2 (see https://anonymous.4open.science/r/RobustZero-SupportMaterials-8512/Tables/Table%20S-2.png). Please note that the number of samples per episode in RobustZero differs from those used in the two baselines (refer to Table 6 in Appendix C.6). Therefore, while the numbers of samples reported in Table S-2 are similar across methods, they are not exactly the same. From Table S-2, by using similar samples, RobustZero achieves higher rewards compared to ATLA-PPO and PROTECTED, demonstrating its superior sample efficiency.
>
> $\textbf{Response to Method and Evaluation Criteria:}$ Following your suggestion, we have studied RobustZero and all baselines on five Mujoco environments, including Hopper, Walker2d, HalfCheetah, Ant and Humanoid. The results are provided in Table S-1 (see https://anonymous.4open.science/r/RobustZero-SupportMaterials-8512/Tables/Table%20S-1.png). The results show consistently that RobustZero outperforms all baselines at defending against state perturbations. This provides evidence that RobustZero also performs well in general and challenging Mujoco environments.
>
> $\textbf{Response to Experimental Designs or Analyses:}$ To ensure a fair comparison, we follow prior works, i.e., ATLA-PPO and PROTECTED by using ATLA-PPO to obtain a worst-case perturbation policy that is applied to all methods. Regarding robust RL methods, e.g., RobustZero, ATLA-PPO and PROTECTED, the agent is trained by using non-perturbed states and perturbed states information. This enables obtaining consistent policies before and after state perturbations. Regarding S-MuZero-worst, it is an extended baseline, representing S-MuZero trained under the worst-case perturbation policy. Importantly, S-MuZero-worst does not incorporate any robustness mechanism. It is trained solely on perturbed states, without any information about non-perturbed states. This causes S-MuZero-worst to be over-trained under the worst-case perturbation policy, making it highly specialized and adapted to that particular perturbation pattern.  As a result, its performance under the worst-case perturbation policy unusually appears higher than its performance under random or no perturbations as observed in Table 1. This explains why its worst-case reward may appear higher than its random-case reward. Note that S-MuZero-worst is deliberately designed to showcase the effect of over-training under worst-case perturbations, regardless of its natural and other performance. Except for S-MuZero-worst, the worst-case rewards are lower than the corresponding natural and random-case rewards.
>
> $\textbf{Response to Essential References Not Discussed}:$ We would like to address your comments as follows: 1) Following your suggestion, we will add the two references. Specifically, a robust adversarial reinforcement learning method is proposed [1] to jointly train an agent and an adversary, where the agent aims to accomplish the primary task objectives while learning to remain robust against disturbances introduced by the adversary. A recent study [2] proposes the use of variational optimization over worst-case adversary distributions, rather than a single adversary, and trains an agent to maximize the lower quantile of returns to mitigate over-optimism; and 2) In our paper, a worst-case state perturbation refers to an adversarial modification of the agent’s observed state that is carefully crafted to minimize its expected return. While, a random-case state perturbation refers to a stochastic disturbance applied to the agent’s observed state. Formal definitions of both perturbation policies are provided in Definitions 4.2 and 4.3 of our paper (see pages 3-4). Additionally, we would like to note that for reference [2], only the abstract is currently available, and the full version has not yet been released online. As a result, we are unable to determine the specific settings used in that work regarding worst-case and random-case perturbations.
>
> $\textbf{Response to Other Comments or Suggestions:}$ We will change “row” to “column”.
>
> $\textbf{Response to Questions for Authors:}$ S-MuZero can solve the tasks with continuous action spaces. We will add this statement in Section 3.3.

---

### Official Review · Reviewer_yCqT · 2025-03-18

**Overall Recommendation:** 3

**Summary:**

The paper introduces RobustZero, an enhanced MuZero framework designed to be robust against both random-case and worst-case adversarial perturbations. RobustZero dynamically balances data generation between these perturbations and incorporates them directly into online training.

**Claims And Evidence:**

The claims presented in the paper are supported by extensive empirical evidence.

**Essential References Not Discussed:**

Given that the paper integrates contrastive representation learning with MuZero, it should further discuss representation learning approaches within the Related Work section, notably EfficientZero (https://arxiv.org/pdf/2111.00210).

**Experimental Designs Or Analyses:**

Consider including training curves (e.g., reward vs. training steps plots) to illustrate learning dynamics and convergence.

**Methods And Evaluation Criteria:**

I suggest employing more general and challenging benchmarks (e.g., board games, MuJoCo) that are standard in the RL community. Pendulum and CartPole appear overly simplistic. Additionally, detailed explanations of the transportation tasks would help clarify their complexity.

**Other Comments Or Suggestions:**

N/A

**Other Strengths And Weaknesses:**

N/A

**Questions For Authors:**

[Q1] How does RobustZero perform on more complex tasks, such as board games (e.g. Go, Chess, Shogi) and high-dimensional continuous control environments (e.g., MuJoCo Humanoid)?

[Q2] The current method involves numerous hyperparameters and seemingly ad-hoc adjustments. Can you reduce the complexity or provide theoretical justification for these choices?

**Relation To Broader Scientific Literature:**

The paper proposes a robust framework addressing sensory noise and adversarial perturbations. Demonstrating effectiveness on more complex tasks could significantly strengthen the proposed method’s relevance. Additionally, the current design involves numerous hyperparameters and ad-hoc choices. Reducing these or providing theoretical justifications would enhance the rigor of the method.

**Theoretical Claims:**

N/A

---

> ### Author Rebuttal · Authors · 2025-03-31
>
> We appreciate your comments and our responses are detailed below.
>
> $\textbf{Response to Method and Evaluation Criteria and Q1}$: We would like to address your comments as follows:
> 1) We have studied RobustZero and all baselines on five Mujoco environments, including Hopper, Walker2d, HalfCheetah, Ant and Humanoid. The results are provided in Table S-1 (see https://anonymous.4open.science/r/RobustZero-SupportMaterials-8512/Tables/Table%20S-1.png). RobustZero consistently outperforms all baselines at defending against state perturbations. This provides evidence that RobustZero also performs well in general and challenging Mujoco environments.
> 2) Following ATLA-PPO and PROTECTED, we adopt an $l_{p}$-norm perturbation model to generate small, semantically invariant perturbations. However, in board games, the states are discrete and highly structured, and even small $l_{p}$-norm perturbations can result in invalid or illegal states that violate game rules. Therefore, this perturbation model is not suitable for board games, and we do not evaluate all methods in such environments.
>  3) We will add the following explanations for the energy and transportation tasks. The three transportation environments support the testing of autonomous driving tasks. Therein, an autonomous driving car interacts with other vehicles to navigate different scenarios: i) Highway$-$Drive fast, avoid collisions, and stay in the right-most lane; ii) Intersection$-$Cross safely, follow traffic rules, and keep a steady speed; and iii) Racetrack$-$Finish quickly while staying on track and driving smoothly. The action space of an autonomous driving car is two-dimensional. The three energy environments support the testing of voltage control tasks with the objective of minimizing the total cost of voltage violations, control errors, and power losses, while meeting both networked and device constraints. The action spaces of IEEE 34-bus, IEEE 123-bus and IEEE 8500-node are 10-dimensional (8 continuous and 2 discrete), 15-dimensional (11 continuous and 4 discrete), and 32-dimensional (22 continuous and 10 discrete), respectively. Thus, these energy environments are complex and high-dimensional.
>
> $\textbf{Response to Experimental Designs or Analyses}$: We have added these training curves (see https://anonymous.4open.science/r/RobustZero-SupportMaterials-8512/Figures/Figure%20S-1.png).
>
> $\textbf{Response to Q2}$:  Our method involves four parameters: $w1$, $w2$, $\lambda_1$, and $\lambda_2$. We clarify their design and selection: 1) One of our contributions is the development of an adaptive mechanism to adjust $w1$  and $w2$, eliminating the need for ad-hoc adjustments. Note that in Appendix C.3, we show the selection of $w2$ across different environments. This is used in ablation studies, where we intentionally fix $w2$ and find best value to isolate its effect and demonstrate the advantage of the adaptive mechanism; 2) $\lambda_1$ adjusts the trade-off between robustness to worst-case perturbations and random perturbations. A larger $\lambda_1$ emphasizes worst-case robustness, while a smaller value favors random-case robustness. When both are considered equally important, setting $\lambda_1=1$ (as done in our experiments) is a reasonable and effective default. Thus, $\lambda_1$ does not require tuning unless specific emphasis is desired; and 3) Among the four parameters, $\lambda_2$ is the only one that requires manual tuning. To keep the selection process simple, we employ a standard grid search. Appendix C.7 analyzes the effect of different $\lambda_2$ values, which is not the selection process. In summary, our method does not rely on ad-hoc adjustments. The adaptive mechanism removes the need to tune $w1$ and $w2$. $\lambda_1$ can be fixed to a default value, e.g., 1. Only $\lambda_2$  requires simple tuning via grid search. Therefore, the hyperparameter selection is not complex.
>
> $\textbf{Response to Essential References Not Discussed}$: We will add the following statements. The contrastive representation learning has been used to improve the sample efficiency [A, B] and enhance the representation ability of states [C], rewards [D], and value functions [E]. Among them, one notable work is EfficientZero [B], which significantly improves the sample efficiency of the MuZero method while maintaining superior performance. It achieves this by using contrastive representation learning to build a consistent environment model and using the learned model to correct off-policy value targets. Different from these studies, we aim to leverage contrastive representation learning to improve the robustness of MuZero-class methods to state perturbations.
>
> [A] Data-efficient reinforcement learning with self-predictive representations
>
> [B] Mastering Atari games with limited data
>
> [C] Planning with goal-conditioned policies
>
> [D] Beyond reward: Offline preference-guided policy optimization
>
> [E] Contrastive learning as goal-conditioned reinforcement learning

---

> > ### Comment · Reviewer_yCqT · 2025-04-02
> >
> > Thanks to the authors for the detailed response.
> >
> > My main concern is still the difficulty of the benchmarks and the performance of the proposed algorithm. On MuJoCo, the performance improvement does not seem to be significant, and in some environments RobustZero is worse than the best baseline, while the better ones are within one standard deviation.

---

> > > ### Author Response · Authors · 2025-04-02
> > >
> > > We sincerely apologize for the insufficient explanation, which may have led to misunderstandings due to space limitations. We would like to take this opportunity to clarify the effectiveness of the proposed RobustZero on the MuJoCo environments as follows.
> > >
> > > 1)	$\textbf{Response to “in some environments RobustZero is worse than the best baseline”.}$  As shown in Table S-1 (see https://anonymous.4open.science/r/RobustZero-SupportMaterials-8512/Tables/Table%20S-1.png), we report natural rewards, worst-case rewards, and random-case rewards across the five MuJoCo environments. First, RobustZero and S-MuZero (i.e., the version of RobustZero without any defense strategies against state perturbations) achieve higher natural rewards than other baselines. Notably, S-MuZero obtains slightly higher natural rewards than RobustZero. This is because S-MuZero is trained without state perturbations. It thus obtains the best natural rewards. However, its worst-case and random-case rewards decrease notably. In comparison, RobustZero can still obtain comparable natural rewards but much better worst-case and random-case rewards. We have also provided an explanation for why S-MuZero slightly outperforms RobustZero in natural rewards on the previously selected eight environments (see Column 1, Lines 381–382, and Column 2, Lines 348–352 on Page 7 of our paper). The exception is on CartPole, where four methods are able to obtain the optimal natural reward as noted in line 661 on page 13 of our paper. Second, RobustZero consistently achieves higher worst-case and random-case rewards than all baselines across all the five MuJoCo environments, further validating its robustness to state perturbations. In summary, RobustZero outperforms all baselines in terms of worst-case and random-case performance, while maintaining comparable natural rewards to S-MuZero across the five MuJoCo environments. These findings are consistent with the results on the previously selected eight environments.
> > >
> > > 2)	$\textbf{Response to the “the better ones are within one standard deviation”.} $  We consider five baselines: ATLA-PPO, PROTECTED, S-MuZero, S-MuZero-worst, and S-MuZero-random. Among them, ATLA-PPO and PROTECTED are model-free DRL methods, while RobustZero, S-MuZero, S-MuZero-worst, and S-MuZero-random are MuZero-class methods. Our key contribution is to propose RobustZero, the first MuZero-class method that is robust to both worst-case and random-case state perturbations. As shown in Table S-1 (see https://anonymous.4open.science/r/RobustZero-SupportMaterials-8512/Tables/Table%20S-1.png), RobustZero achieves the best worst-case and random-case rewards. Importantly, there are no overlapping standard deviations between RobustZero and the other three MuZero-class baselines (S-MuZero, S-MuZero-worst, and S-MuZero-random) across all five MuJoCo environments. This demonstrates that the performance improvements brought by RobustZero in robustness are statistically significant within the MuZero family. In addition, although there are overlapping standard deviations between RobustZero and the model-free baselines (ATLA-PPO and PROTECTED) in some cases, RobustZero still achieves the highest average performance.
> > > 3)	$\textbf{Response to “On MuJoCo, the performance improvement does not seem to be significant”. }$ As discussed above, RobustZero significantly improves the robustness of MuZero-class methods to both worst-case and random-case state perturbations, while maintaining high natural performance across the five MuJoCo environments.  Thus, RobustZero performs well in general and challenging MuJoCo environments.
> > >
> > > We hope these clarifications address your concerns.

---

### Decision · Program_Chairs · 2025-05-01

**Decision:**

Accept (poster)

**Comment:**

This paper proposes RobustZero, a MuZero-class method that is robust to worst-case and random-case state perturbations, with zero prior knowledge of the environment’s dynamics. Although reviewers show some concerns, particularly for the difficulty of the benchmarks and the performance of the proposed algorithm, all reviewers finally voted for strong/weak accepts. Thus, we are happy to accept this paper. However, please make sure to address the reviewer concerns in the final version.